# Backdoor Mitigation via Invertible Pruning Masks

**Kealan Dunnett**[1], **Reza Arablouei**[2], **Dimity Miller**[1], **Volkan Dedeoglu**[2], and **Raja Jurdak**[1]

[1]Queensland University of Technology, Brisbane Australia
[2]Data61, CSIRO, Pullenvale QLD, Australia

## Abstract

Model pruning has gained traction as a promising defense strategy against backdoor attacks in deep learning. However, existing pruning-based approaches often fall short in accurately identifying and removing the specific parameters responsible for inducing backdoor behaviors. Despite the dominance of fine-tuning-based defenses in recent literature, largely due to their superior performance, pruning remains a compelling alternative, offering greater interpretability and improved robustness in low-data regimes. In this paper, we propose a novel pruning approach featuring a learned *selection* mechanism to identify parameters critical to both main and backdoor tasks, along with an *invertible* pruning mask designed to simultaneously achieve two complementary goals: eliminating the backdoor task while preserving it through the inverse mask. We formulate this as a bi-level optimization problem that jointly learns selection variables, a sparse invertible mask, and sample-specific backdoor perturbations derived from clean data. The inner problem synthesizes candidate triggers using the inverse mask, while the outer problem refines the mask to suppress backdoor behavior without impairing clean-task accuracy. Extensive experiments demonstrate that our approach outperforms existing pruning-based backdoor mitigation approaches, maintains strong performance under limited data conditions, and achieves competitive results compared to state-of-the-art fine-tuning approaches. Notably, the proposed approach is particularly effective in restoring correct predictions for compromised samples after successful backdoor mitigation.

## 1   Introduction

The widespread adoption of deep learning across various sectors has brought increased scrutiny to its associated security vulnerabilities. While the performance benefits of modern deep learning methods are well recognized, their inherent opacity makes them susceptible to various forms of exploitation [1]. Notable examples of such threats are adversarial [2] and data poisoning [3] attacks. In classification tasks, backdoor attacks are a critical concern. First introduced by [4], these attacks compromise model integrity by injecting a hidden behavior: inputs containing a specific trigger are misclassified into a target class, while clean inputs (i.e., those without this trigger) remain correctly classified. This allows the adversary to embed malicious functionality without disrupting the model's performance on benign data.

The potential impact of backdoor attacks is especially severe in safety-critical applications that rely on automated decision-making. For example, driver assistance systems often depend on image classification modules to detect critical objects such as traffic signs. A subtle physical modification, such as placing a sticker on a stop sign, may exploit a backdoored model to induce targeted misclassification, with catastrophic consequences [5]. These risks are exacerbated in scenarios where model training is outsourced to third parties. A recent survey by [6] revealed that 48.1% of respondents rely on third-party models, typically fine-tuning them for their specific use cases.

39th Conference on Neural Information Processing Systems (NeurIPS 2025)

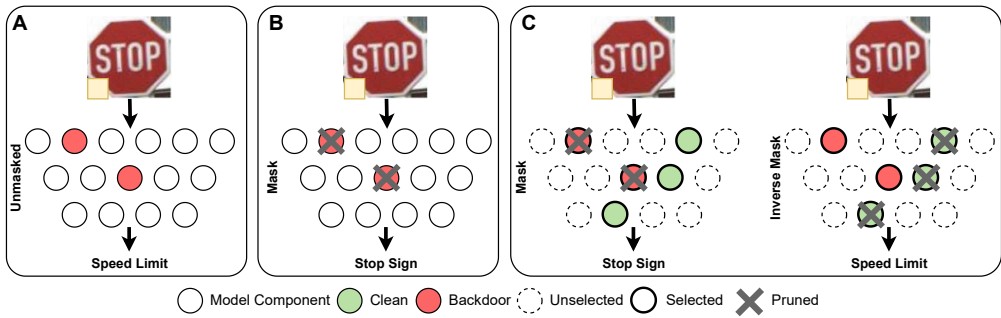

Figure 1: The unpruned backdoored model (A), conventional model pruning (B), and model pruning via an invertible mask (C).

Backdoor mitigation aims to remove the malicious functionality induced by such attacks while preserving the model's original classification capabilities. Within image classification, this has proven particularly challenging, as demonstrated by the recent studies [7, 8]. Existing backdoor mitigation approaches generally fall into two broad categories: model pruning and fine-tuning [7]. Pruning-based approaches seek to identify and remove model components (e.g., convolutional filters) associated with the backdoor behavior, whereas fine-tuning-based approaches adjust model parameters through a retraining process. To date, fine-tuning approaches have generally outperformed pruning approaches [8, 7], establishing fine-tuning as the dominant paradigm in backdoor mitigation literature.

However, it has been observed in [8] that the performance of fine-tuning approaches degrade more significantly when limited data is available. Moreover, although fine-tuning approaches exhibit robust capabilities in removing backdoor behaviors, their ability to restore correct classification of backdoor-triggered inputs remains a persistent challenge [7, 8]. To address these limitations, we introduce a novel pruning-based approach to backdoor attack mitigation that aims to narrow the performance gap between fine-tuning and pruning. Our core contribution lies in rethinking the pruning paradigm through the introduction of two new principles: mask invertibility and component selection.

Current pruning-based approaches solely focus on removing model components associated with the backdoor task. While this perspective is essential, relying on it in isolation yields inconsistent and often brittle performance across different attack scenarios [7]. In contrast, we propose using an invertible mask that unifies the standard pruning perspective with a complementary one to enhance robustness: selectively removing *clean* components while *retaining* those related to the backdoor. Specifically, we evaluate model behavior under two opposing conditions: (i) pruning clean-task components (via the inverse mask) and (ii) pruning backdoor-task components (via the standard mask). This dual-view approach enables a more principled dissection of model components, revealing their respective roles in clean and backdoor tasks. To ensure the invertible mask remains practical and avoids excessive pruning, we introduce a selection mechanism that identifies components highly relevant to both the clean and backdoor tasks, marking them as *eligible* for pruning. Subsequently, components not selected by this mechanism are automatically preserved and excluded from both pruning processes. Fig. 1 illustrates the key differences between our approach (C) and traditional pruning (B).

To realize this idea, we develop a bi-level optimization framework inspired by SAU [9]. Similar to SAU, we address the lack of ground-truth backdoor data by synthesizing adversarial examples, additively perturbed versions of clean inputs that are misclassified in the same way by both the original model and a pruned variant obtained using the inverse mask (i.e., with clean components removed and backdoor components retained). Subsequently, we estimate a pruning mask aimed at suppressing the influence of these synthesized triggers. In contrast to SAU, our approach replaces fine-tuning with pruning for both trigger synthesis and backdoor mitigation. Accordingly, we reformulate the objective to move beyond incremental parameter adaptation, focusing instead on structurally disentangling the model components associated with backdoor and clean behaviors. This enables a more robust and interpretable mechanism for jointly identifying and mitigating backdoor behaviors.

Through extensive experiments, we demonstrate that our approach significantly outperforms existing pruning-based defenses. Additionally, we show that it remains robust with limited data, highlighting

its practicality and generalizability in real-world scenarios. Compared to its fine-tuning-based contenders, our approach more effectively restores correct classifications of backdoor samples following mitigation. This suggests that pruning, when augmented with mask invertibiity and component selection, can offer a more refined and resilient solution for backdoor mitigation.

## 2 Related Work

**Backdoor Attacks:** The seminal BadNets attack [4] demonstrated how embedding a backdoor task into a model could be achieved by manipulating a subset of training data, such as adding a colored square to specific input samples. Building on this foundational concept, subsequent research has led to increasingly sophisticated backdoor attacks. For example, [10] introduce an encoder-decoder-based approach that generates sample-specific triggers by jointly optimizing the encoder-decoder network and the classification model. Beyond trigger-based techniques, alternative strategies such as relabeling attacks have also been explored. For instance, [11] present an all-to-all attack that disrupts multiple classes simultaneously. In this work, we focus on the targeted attack setting, where the objective is to consistently misclassify samples toward a fixed target class $t$.

**Backdoor Defense:** Mitigating backdoor attacks has predominantly involved model pruning and fine-tuning using clean training data, as initially proposed in [12]. Subsequent research, such as [13, 14], has further refined these techniques. Among pruning-based approaches, ANP [15], AWN [16], and RNP [17] closely align with our approach as ANP and AWN employ bi-level optimization to estimate optimal masks for channel-wise pruning of convolutional layers. Recent approaches, such as FMP [18] and NFT [19], utilize adversarial examples and the MixUp protocol to learn pruning masks that effectively suppress backdoors. FT-SAM [20] and SAU [9], state-of-the-art fine-tuning approaches, mitigate backdoors by utilizing sharpness-aware minimax optimization and modified adversarial training via shared examples, respectively. Moreover, BTI-DBF [21] trains a generative model via initial pruning to generate samples for fine-tuning, then applies an inverse-like binary mask to maximize the classification error on clean samples as part of its mitigation strategy. For comprehensive benchmarks and comparative analyses of these approaches, we refer readers to [7, 8].

While existing pruning approaches primarily focus on mask optimization (e.g., ANP, AWN, and RNP), and fine-tuning approaches solely rely on parameter adaptation (e.g., FT-SAM and SAU), our approach bridges these strategies, using mask invertibility and component selection to systematically disentangle backdoor components from clean behavior, achieving robust mitigation even with limited data.

## 3 Preliminaries

**Model Pruning:** Channel-wise masking of convolutional neural networks has been widely employed in backdoor mitigation literature [15–17, 19, 18]. Pruning-based approaches operate on the premise that specific subsets of convolutional components are responsible for backdoor behaviours. Empirical evidence supports this assumption [22, 16, 12], motivating the use of channel-wise masking to isolate and eliminate these deleterious components. This approach involves channel-wise multiplication of the kernel weights of each convolutional layer by a set of learned parameters bounded between 0 and 1, denoted as the mask vector $\mathbf{a}$.

When an entry of $\mathbf{a}$ is zero, the corresponding channel in the kernel tensor is effectively pruned through the application of the mask. However, in practice, most model pruning approaches do not directly apply masks to model parameters. Instead, $\mathbf{a}$ is commonly applied to the activation maps produced by each layer during the forward pass. Given the model parameters $\theta$ and the set of pruning masks $\mathcal{A} = \{\mathbf{a}_1, \cdots, \mathbf{a}_N\}$ for $N$ convolutional layers, we represent the masked model as $\theta \odot \mathcal{A}$, where $\odot$ denotes the channel-wise application of each mask in $\mathcal{A}$ to its corresponding layer in $\theta$.

**Threat Model:** In a targeted attack scenario, the adversary embeds a backdoor task into the model, causing inputs $\hat{x}$ containing a specific trigger pattern $\rho$ to be classified as a designated target class $t$. This is achieved by training the model using two datasets: the clean dataset $\mathcal{D}_c = \{(x, y)\}$, comprising original inputs $x$ with correct labels $y$, and the backdoor dataset $\mathcal{D}_b = \{(\hat{x}, t)\}$. In certain

cases, such as in [23], $\rho$ is dynamic, varying for each $\hat{x}$. The adversary's goal is to train a set of model parameters $\theta'$ that correctly classify both $D_c$ and $D_b$.

We consider a range of attacks documented in the literature without imposing any additional constraint beyond those outlined in the original works. For instance, in extreme cases like in [24], the attacker is assumed to have complete control over the training process. From the defender's perspective, we assume access to a limited set of correctly labelled clean inputs $\mathcal{D}_m = \{(x, y)\}$ (i.e., inputs with no backdoor trigger) and the backdoored model parameters $\theta'$. In practice, the reliance on outsourced training often restricts defenders to a limited set of clean samples, underscoring the need for mitigation strategies that remain affective under such constraints.

Furthermore, while our evaluation primarily focuses on convolutional neural networks, we also explore how IMS can be adapted for use with transformer-based architectures. Given the growing significance of Vision Transformers (ViTs) and the lack of existing pruning-based defenses tailored to them, we provide preliminary results demonstrating that ViTs can be successfully pruned using IMS to mitigate backdoor behaviour.

## 4 Proposed Approach

In this section, we introduce *Invertible Masking using Selection* (IMS), a novel model-pruning-based approach for backdoor mitigation. We begin by defining the characteristics of an invertible mask and elucidating the importance of component selection. We then describe the proposed IMS approach within the context established in section 3.

### 4.1 Invertible Masking using Selection

**Invertibility:** To overcome the limitations of existing pruning-based methods, we introduce a more flexible pruning paradigm termed invertibility, aimed at more effectively distinguishing between model components critical to backdoor and clean tasks. Unlike conventional pruning (e.g., ANP, AWN, and RNP), which is treated as a one-way subtractive process focused solely on removing backdoor components, we leverage invertibility to contrast the model's behaviour under two complementary pruning configurations: one in which clean components are pruned while backdoor components are retained (using the inverse mask), and another where the standard pruning mask is applied. This dual-objective approach serves two purposes: it verifies the correct identification of critical clean components, preventing their removal, and assesses whether the backdoor behavior persists when only backdoor components are retained and the clean components are pruned. Thus, enforcing mask invertibility enables a more nuanced distinction between components associated with clean and backdoor behaviors.

**Selection:** Existing approaches generally categorize model components as being associated with either the backdoor task or the clean task. In contrast, we relax this binary assumption through *selection*, allowing most model components to be shared between the two pruned models generated by applying the mask and its inverse. Specifically, we identify a subset of model components that are most relevant to both backdoor and clean tasks and apply the mask and its inverse exclusively to these selected components. The remaining unselected components form a shared backbone that supports both backdoor and clean tasks and are thus excluded from pruning by both the mask and its inverse, as depicted in Fig. 1 (C). Consequently, the selection mechanism helps prevent over-pruning and enables the practical construction of the mask and its inverse. In the absence of shared components, the mask and its binary complement must partition the model into two connectivity-preserving subnetworks without entirely pruning any layer. However, this requirement conflicts with the objective of sparsity, imposing a prohibitive constraint.

**Selective Invertible Masking:** To extend channel-wise masking to support invertibility through selection, we introduce a learned selection parameter vector $\mathbf{s} \in [0, 1]^{C_{\text{out}}}$ for each layer $l$. This selection vector, combined with the preliminary pruning mask $\mathbf{a} \in [0, 1]^{C_{\text{out}}}$ (introduced in section 3), defines the mask $\mathbf{a}'$ and its inverse $\bar{\mathbf{a}}'$ for the $l^{\text{th}}$ layer as

$$\mathbf{a}' = \sigma(k\,[\mathbf{a} - 0.5]) + \mathbf{s} \circ \sigma(k[(1 - \mathbf{a}) - 0.5]) \tag{1}$$

$$\bar{\mathbf{a}}' = \sigma(k[(1 - \mathbf{a}) - 0.5]) + \mathbf{s} \circ \sigma(k\,[\mathbf{a} - 0.5]) \tag{2}$$

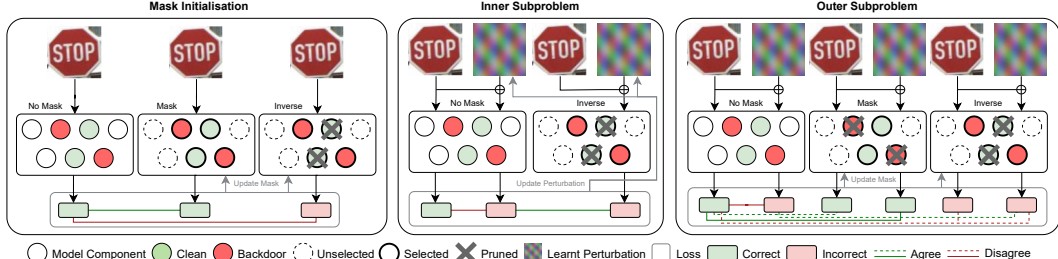

Figure 2: Summary of the bi-level optimization framework, illustrating the agreement and disagreement loss terms and their role in updating the mask or perturbation for each described subproblem. Dashed lines in the Outer Subproblem diagram indicate objectives shared with Mask Initialisation and Inner Subproblem.

where $\sigma(\cdot)$ represents the sigmoid function applied element-wise, $k$ is a scaling parameter that refines channel selection, $\circ$ denotes element-wise multiplication, and addition/subtraction is performed element-wise. Note that, $\bar{\mathbf{a}}' \neq 1 - \mathbf{a}' \neq 1 - \mathbf{a}$, when $\mathbf{s} > 0$ (element-wise), meaning the inverse mask is not simply the binary complement mask, as is the case in BTI-DBF [21].

In Fig. 7 (Appendix A), we visualize the relationship between $\mathbf{a}'$ and $\bar{\mathbf{a}}'$ across various $\mathbf{a}$ and $\mathbf{s}$ values. Moreover, we also provide a theoretical analysis of the invertibility of the proposed mask in Appendix A. Shared components (Unselected in Fig. 1) correspond to entries of $\mathbf{s}$ with values near one, while selected components, which may be pruned by either the mask or its inverse (Selected in Fig. 1) have corresponding entries in $\mathbf{s}$ with values close to zero. For the selected parameters, the mask and its inverse function as opposing counterparts, each targeting complementary subsets (one retaining what the other prunes), as illustrated in Fig. 7.

## 4.2 Optimization Framework

To estimate the optimal values of $\mathcal{A}$ and $\mathcal{S} = \mathbf{s}_1, \cdots, \mathbf{s}_N$, we minimize a composite objective within a bi-level optimization framework. The inner subproblem synthesizes sample-specific perturbations $\delta$ for clean inputs $x \in \mathcal{D}_m$ using the inverse mask $\bar{\mathcal{A}}' = \bar{\mathbf{a}}'_1, \cdots, \bar{\mathbf{a}}'_N$, capturing the dynamic nature of potential backdoor triggers. As the defender lacks access to actual backdoor data, these perturbations act as proxies for the adversarial manipulations. The outer subproblem then updates $\mathcal{A}$ and $\mathcal{S}$ to suppress backdoor behaviors while preserving clean-task functionality. During the initialization phase, we estimate $\mathcal{A}$ and $\mathcal{S}$ to identify components crucial only for the clean task. We provide further details of the optimization procedure in the subsequent sections, and an overview in Fig. 2. Moreover, in Appendix B, we summarize the training procedure of IMS.

**Model Outputs:** Given model parameters $\theta$, the model mask $\mathcal{A}' = \{\mathbf{a}'_1, \cdots, \mathbf{a}'_N\}$ and its inverse $\bar{\mathcal{A}}'$, the softmax model outputs [denoted using the function $f(\cdot)$] for a clean input $x$ and its perturbed version $\hat{x} = x + \delta$ are expressed as

$$
\begin{aligned}
p &= f(x, \theta), \ p_{\mathcal{A}'} = f(x, \theta \odot \mathcal{A}'), \ \hat{p}_{\mathcal{A}'} = f(\hat{x}, \theta \odot \mathcal{A}'), \\
\hat{p} &= f(\hat{x}, \theta), \ p_{\bar{\mathcal{A}}'} = f(x, \theta \odot \bar{\mathcal{A}}'), \ \hat{p}_{\bar{\mathcal{A}}'} = f(\hat{x}, \theta \odot \bar{\mathcal{A}}').
\end{aligned}
\tag{3}
$$

**Loss Functions:** The optimization framework incorporates agreement and disagreement loss functions, defined for given softmax outputs $\hat{q}$ and $q$ as $\mathcal{L}_{\text{Disagree}}(\hat{q}, q) = -\log(1 - \langle \hat{q}, q \rangle)$ and $\mathcal{L}_{\text{Agree}}(\hat{q}, q) = -\log(\langle \hat{q}, q \rangle)$, where $\langle \hat{q}, q \rangle = \sum_{c=1}^{C} \hat{q}_c \cdot q_c$ denotes the dot product and $C$ is the number of classes. Minimizing the agreement loss promotes alignment between model outputs, while minimizing the disagreement loss encourages their divergence.

**Mask Initialisation:** To estimate the initial values of the mask and its inverse, we use the clean data to identify the model components critical solely to the clean task. As shown in Fig. 2, this involves updating $\mathcal{A}$ and $\mathcal{S}$ such that $p$ aligns with $p_{\mathcal{A}'}$ while diverging from $p_{\bar{\mathcal{A}}'}$. Thus, we formulate the initialization optimisation problem as

$$
\min_{\mathcal{S}, \mathcal{A}} \ \mathcal{L}_{\text{Agree}}(p_{\mathcal{A}'}, p) + \mathcal{L}_{\text{Disagree}}(p_{\bar{\mathcal{A}}'}, p) + \frac{\lambda}{|\mathcal{S}|} \|\mathcal{S}\|_1.
\tag{4}
$$

The last term is a regularization penalty, defined as the $\ell_1$ norm of $\mathcal{S}$ scaled by the parameter $\lambda$ and normalised by the number of its entries $|\mathcal{S}|$. This term encourages sparsity in $\mathcal{S}$, mitigating the risk of excessive pruning. By initializing the mask such that it preserves correct classification of clean data while its inverse leads to their misclassification, we compel the inner subproblem to rely on the unselected and backdoor components to estimate the backdoor trigger.

**Inner subproblem:** To synthesize perturbed inputs $\hat{x}$ that elicit the backdoor behavior, we estimate $\delta$ for all available clean data instances by utilizing the outputs of the unmasked and inverse-masked models. Specifically, we aim to encourage disagreement between $\hat{p}$ and $p$ while simultaneously promoting agreement between $\hat{p}$ and $\hat{p}_{\bar{\mathcal{A}}'}$, as illustrated in Fig. 2. Since the inverse mask prunes components critical to the clean task, this objective drives the estimation of perturbations to rely on components not associated with the clean task. Therefore, this effectively biases perturbations towards utilising backdoor components to induce misclassification, a phenomenon supported by our experimental results. We express the inner subproblem as

$$\min_{\|\delta\|_\infty < \epsilon} \mathcal{L}_{\text{Disagree}}(\hat{p}, p) + \mathcal{L}_{\text{Agree}}(\hat{p}, \hat{p}_{\bar{\mathcal{A}}'}), \tag{5}$$

where the $\ell_\infty$ norm constraint $\|\delta\|_\infty < \epsilon$ restricts perturbations to remain bounded within a specified tolerance. We set $\epsilon = 1$ in all experiments. Note that, due to the use of the AdamW optimizer rather than signed gradients, $\epsilon = 1$ serves as a hard upper bound on the perturbation magnitude rather than a target value.

**Outer subproblem:** Using the estimated perturbations $\delta$, the outer subproblem optimizes $\mathcal{S}$ and $\mathcal{A}$ to mitigate the effect of $\delta$. Specifically, we design it to minimize the agreement loss for $\tilde{p}_{\mathcal{A}'}$ and $p$, thereby restoring the correct classification of $\hat{x}$ when the mask $\mathcal{A}$ is applied. To maintain consistency with the objectives of the mask initialization and the inner subproblem, we incorporate their respective loss terms into the outer subproblem's objective function, as indicated by the dashed lines in Fig. 2. Their integration ensures that the updated $\mathcal{S}$ and $\mathcal{A}$ effectively mitigate the impact of the perturbations during inference, without compromising the objectives established in the preceding steps. Consequently, we define the outer optimization subproblem as

$$\min_{\mathcal{S}, \mathcal{A}} \mathcal{L} + \frac{\lambda}{|\mathcal{S}|} \|\mathcal{S}\|_1, \tag{6}$$

$$\mathcal{L} = \mathcal{L}_{\text{Agree}}(p_{\mathcal{A}'}, p) + \mathcal{L}_{\text{Agree}}(\tilde{p}_{\mathcal{A}'}, p) + \mathcal{L}_{\text{Disagree}}(\hat{p}_{\bar{\mathcal{A}}'}, p) + \mathcal{L}_{\text{Agree}}(\hat{p}, \hat{p}_{\bar{\mathcal{A}}'}) + \mathcal{L}_{\text{Disagree}}(p_{\bar{\mathcal{A}}'}, p). \tag{7}$$

We solve the outer subproblem by initially setting $\lambda = 0$ and then increasing it to $\lambda = 10$.

## 5 Evaluation

In this section, we evaluate the effectiveness of the proposed IMS approach in mitigating backdoor attacks within the BackdoorBench [7] evaluation tool. Adopting the benchmarking methodology of [8], we conduct a comprehensive evaluation across diverse experimental settings. These include 8 backdoor attacks, 4 model architectures, 3 datasets, and 3 poisoning ratios, a total of 288 distinct test cases.

### 5.1 Experimental Setup

We consider a range of representative backdoor attacks, including BadNets [4], Blended [25], Signal [26], LF [27], SSBA [10], IAB [23], BPP [24], and WaNet [28]. We use the default attack configurations, as specified by *BackdoorBench* [7], with poisoning ratios of 1%, 5%, and 10%. In each case, the first class (indexed by 0 in Python) is the backdoor attack target class. We utilize the CIFAR-10, German Traffic Sign Recognition Benchmark (GTSRB), and Tiny-ImageNet datasets, which contain 10, 43, and 200 classes, respectively. Moreover, we consider the PreAct-ResNet18 (ResNet), VGG-19 with batch normalisation (VGG), EfficientNet-B3 (EfficientNet), and MobileNetV3-Large (MobileNet) model architectures. For each dataset, we consider three data settings based on sample per class (SPC) values of 2, 10 and, 100.

We benchmark IMS against ANP [15], ANW [16] and RNP [17], given their methodological alignment with our approach. Additionally, we include FP [12] as a foundational baseline and the more

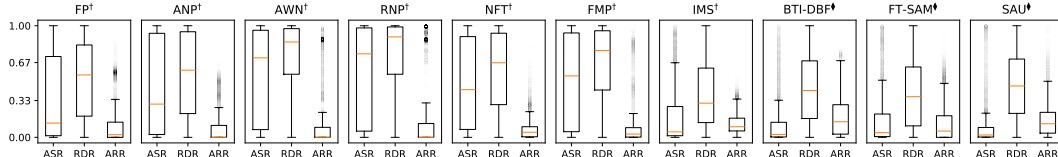

Figure 3: Box plots illustrating ASR, RDR, and ARR results of IMS against various pruning and fine-tuning approaches across all tested settings. †: Pruning, ♦: Fine-tuning.

recent related approaches NFT [19] and FMP [18]. To provide a broader perspective, we further evaluate IMS against state-of-the-art fine-tuning-based approaches FT-SAM [20] and SAU [9]. We also include BTI-DBF [21] due to its use of an inverse-like mask. We provide the implementation of IMS on GitHub.[1]

**Performance Measures:**   To assess the effectiveness of backdoor mitigation, we adopt three performance measures as defined in [8]:

- *Accuracy reduction ratio* (ARR), which quantifies the impact of mitigation on the original classification task. Lower ARR values indicate minimal disruption to the model's primary functionality.
- *Attack success ratio* (ASR), which measures the extent to which the backdoor trigger continues to induce misclassifications after mitigation. An effective mitigation strategy results in a low ASR.
- *Recovery difference ratio* (RDR), which evaluates how effectively the mitigation restores correct classification of samples containing the backdoor trigger. A low RDR signifies better recovery from backdoor contamination.

Ideally, all three measures (ARR, ASR, and RDR) approach zero, indicating minimal impact of mitigation on the original task (low ARR), effective neutralization of backdoor behavior (low ASR), and accurate classification of backdoored samples (low RDR). We provide detailed mathematical formulations of these measures in Appendix C. Moreover, in Appendix K, we extend the experimental analysis presented in section 5.2 with supplementary plots and discussions. We further present additional evaluations of IMS, including a computational complexity analysis, performance on an ImageNet subset, the role of mask initalization, the impact of $\lambda$, the effect of $k$, the significance of selection, and robustness against Backdoor Reactivation [29]. These are detailed in Appendices D, E, F, G, H, I, and J, respectively.

## 5.2   Results

In Fig. 3, we provide an overview of the performance of IMS relative to prominent pruning- and fine-tuning-based approaches. We use box plots to summarize the distribution of ASR, RDR, and ARR values across all considered experimental scenarios. The results indicate that IMS consistently outperforms existing pruning approaches, FP, ANP, AWN, RNP, NFT, and FMP, in both ASR and RDR performance. Moreover, IMS demonstrates competitive ASR and improved RDR and ARR performance relative to state-of-the-art fine-tuning approaches, BTI-DBF, FT-SAM, and SAU. In particular, IMS's superior ARR and RDR performance represent a critical advantage, as enhanced RDR , particularly when ARR remains largely unaffected, signifies a defence that not only suppresses the backdoor behaviour and restores correct classification of compromised samples but also preserves clean task performance.

In Fig. 4, we compare the performance of the evaluated approaches across different SPC settings, with the corresponding median values of ASR, RDR, and ARR provided in Table 1.Compared to other pruning-based approaches, IMS consistently achieves higher ASR and RDR performance across all data availability settings, with only a marginal increase in ARR. When benchmarked against fine-tuning-based approaches, IMS demonstrates superior RDR performance in the 2 and 10 SPC settings, accompanied by lower median ARR and reduced variance. In the 100 SPC setting, IMS outperforms BTI-DBF and SAU in terms of RDR and remains competitive with FT-SAM in terms of ARR and RDR. These findings underscore the robustness of IMS in mitigating backdoor attacks, especially under limited data conditions, effectively narrowing the performance gap between pruning- and fine-tuning-based defenses. Importantly, IMS substantially closes the performance gap

---

[1]`https://github.com/WhoDunnett/BackdoorBenchmark`

Table 1: Median ARR, ASR, and RDR results (x100) for IMS and various existing pruning and fine-tuning approaches with different SPC values. Bold values indicate the best performance among pruning or fine-tuning approaches, while underlined values represent the overall best performance.

| SPC | Metric | Pruning | | | | | | | Fine-Tuning | | |
| --- | --- | --- | --- | --- | --- | --- | --- | --- | --- | --- | --- |
| | | FP | ANP | AWN | RNP | NFT | FMP | IMS | BTI-DBF | FT-SAM | SAU |
| 2 | ARR | 3.2 | **0.0** | **0.0** | **0.0** | 8.6 | 5.4 | 17.3 | 26.0 | **19.0** | 23.5 |
| | ASR | 29.5 | 51.4 | 79.0 | 71.1 | 48.6 | 76.7 | **5.6** | 4.6 | 4.6 | **1.9** |
| | RDR | 66.5 | 65.8 | 87.2 | 90.2 | 75.0 | 90.5 | **38.9** | 56.4 | **55.3** | 57.7 |
| 10 | ARR | 6.2 | **0.0** | **0.0** | **0.0** | 3.8 | 3.3 | 9.8 | 14.7 | **3.6** | 11.6 |
| | ASR | 6.0 | 25.2 | 74.7 | 80.6 | 52.0 | 59.0 | **4.7** | 2.4 | 4.8 | **2.0** |
| | RDR | 56.1 | 58.5 | 86.6 | 93.3 | 70.3 | 80.6 | **28.5** | 45.2 | **33.7** | 42.1 |
| 100 | ARR | 0.4 | **0.0** | 0.4 | **0.0** | 1.4 | 0.4 | 6.5 | 7.8 | **0.0** | 4.5 |
| | ASR | 12.4 | 16.7 | 66.5 | 80.6 | 34.5 | 32.0 | **4.2** | **1.9** | 3.3 | 2.3 |
| | RDR | 36.7 | 56.9 | 82.1 | 89.6 | 59.0 | 55.7 | **21.2** | 24.4 | **15.5** | 37.1 |

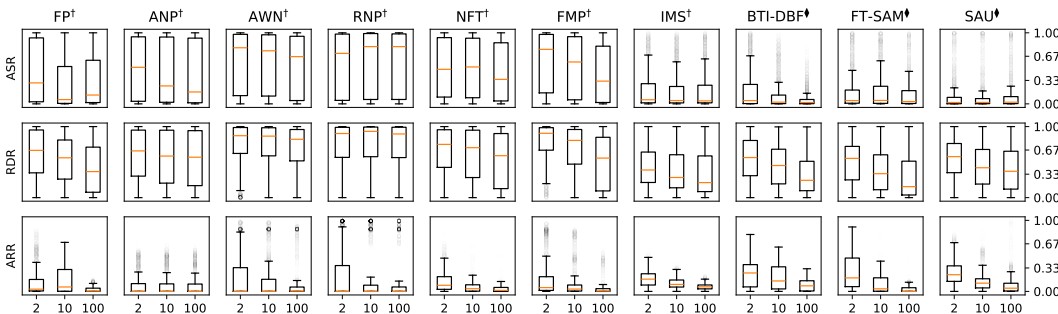

Figure 4: Box plots illustrating ASR, RDR, and ARR results for IMS and various existing pruning and fine-tuning approaches with SPC values of 2, 10, and 100. †: Pruning, ♦: Fine-tuning.

to leading fine-tuning-based approaches, while maintaining the interpretability benefits inherent to pruning-based approaches.

In Fig. 5, we compare the performance of IMS with that of the best-performing pruning approaches from Fig. 3 for various attack types and model architectures. The results in Fig. 5a show that, unlike ANP and NFT, IMS exhibits competitive performance across all considered architectures. Notably, AWN and NFT show large variations in RDR and ASR performance, highlighting the challenges of designing pruning-based approaches that are robust to architectural changes. From Fig. 5b, we observe that IMS is less affected by attack variations compared to other pruning approaches. Specifically, the ASR and RDR performance of ANP and NFT varies considerably under different attack scenarios. While IMS also experiences some performance fluctuations, its overall consistency is substantially higher. In Appendix K.1, we provide a comprehensive comparison of IMS with fine-tuning baselines, revealing that RDR performance patterns of IMS, BTI-DBF, FT-SAM, and SAU are similar across attack types, with IMS exhibiting lower variance and better median RDR performance in most cases. These findings suggest that certain attacks (e.g., Blended, LF, Signal, SSBA) pose similar challenges for both pruning and fine-tuning approaches.

In Fig. 6, we present the scatter plots of the proportion of $\hat{x}$ (the perturbed inputs found by solving the inner subproblem) classified as the target class post mitigation and the ASR achieved by IMS, for each dataset. Note that knowledge of the proportion of $\hat{x}$ classified as the target class is not accessible to the defender during mitigation, as the target class is presumed to be unknown. The results indicate that as the proportion of $\hat{x}$ samples classified as the target class increases, ASR decreases. Moreover, we observe that mandating mask invertibility effectively biases $\hat{x}$ toward the backdoor task, as in most cases, the target class proportion exceeds its expected frequency, as indicated by the dashed lines in Fig. 6. The results support the intuition outlined in Section 4.1, suggesting that the inverse mask directs perturbations to utilize backdoor components. Given that IMS reduces ASR significantly when the target class proportion is high, future work can focus on further intensifying this effect to improve robustness.

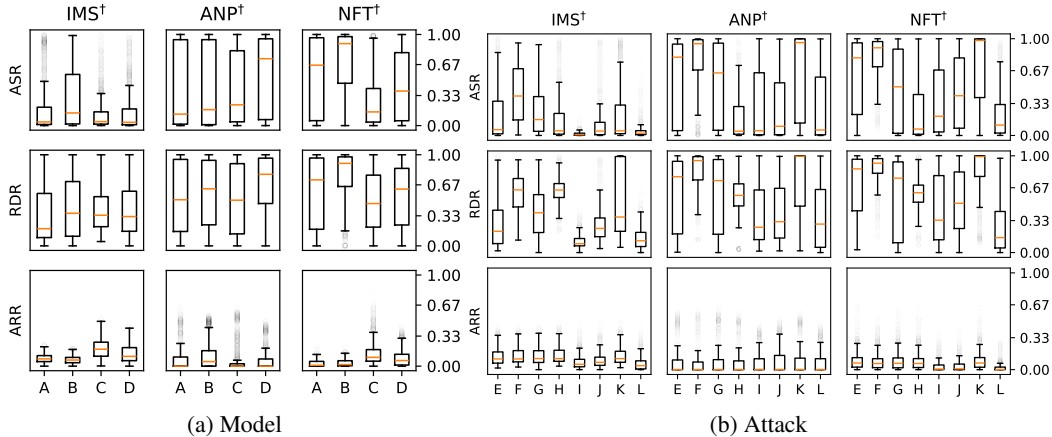

(a) Model                                                    (b) Attack

Figure 5: Box plots illustrating ASR, RDR, and ARR results for IMS, ANP, and NFT, across different model and attack settings. In (A) A: VGG, B: ResNet, C: EfficientNet, D: MobileNet. In (B) E: BadNet, F: Blended, G: LF, H: Signal, I: BPP, J: Inputaware, K: SSBA, L: WaNet.

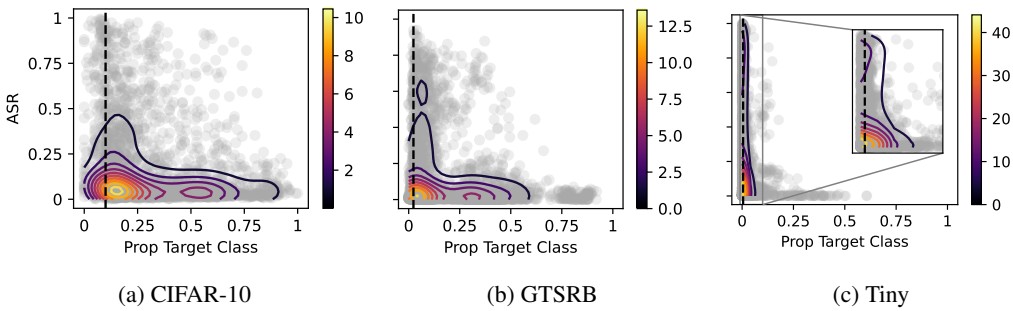

(a) CIFAR-10                          (b) GTSRB                          (c) Tiny

Figure 6: Scatter plots of the proportion of $\hat{x}$ classified as the target class vs ASR of IMS for each dataset. The vertical dashed lines represent the expected frequency of the target class $\frac{1}{C}$, with $C$ being the number of classes, 10, 43, and 200 for CIFAR-10, GTSRB, and Tiny-ImageNet, respectively.

## 5.3   Adapting IMS to ViTs

Vision Transformers (ViTs) have gained significant popularity in image classification tasks in recent years. Similar to CNNs, several studies have shown that ViTs are highly overparameterized and can be pruned to improve efficiency. Notably, in [30], it is demonstrated that ViTs can sustain high levels of pruning while maintaining strong performance. Building on this insight, pruning encoder blocks appears to be a promising strategy for extending pruning-based approaches to ViT architectures. We hypothesize that backdoor behavior in ViTs may be encoded within specific embedding dimensions of each encoder block. Therefore, applying a mask to selectively prune these embedding dimensions represents a natural and effective starting point for backdoor mitigation.

To evaluate this strategy, we apply IMS to ViT-B/16 models trained on GTSRB with the BadNet, BPP, and IAB attacks, using a poisoning ratio of 10%. Table 2 reports the median performance (ARR, RDR, and ASR) of IMS across different SPC values. Since it is observed in [30] that ViTs tolerate substantial pruning, we omit the sparsity regularization term from Equation (6) in this adaptation.

IMS performs strongly on BadNet and BPP, maintaining low ASR, RDR, and ARR (all below 10) even as SPC is reduced. On IAB, IMS is less effective, though modest reductions in ASR and RDR are still observed.

This preliminary investigation highlights the potential of IMS and more broadly pruning-based defenses for application to ViTs. Future work can explore which components within the transformer architecture are most suitable for pruning, as the embedding dimension is only one of several possible targets. For example, the query, key, and value projection matrices may offer promising pruning opportunities, as suggested by recent feature selection methods such as [31].

Table 2: Median ASR, RDR, and ARR of IMS when adopted for ViTs.

| Attack | SPC = 2 | | | SPC = 10 | | | SPC = 100 | | |
|---|---|---|---|---|---|---|---|---|---|
| | ASR | RDR | ARR | ASR | RDR | ARR | ASR | RDR | ARR |
| BadNet | 2.2 | 6.3 | 4.0 | 6.9 | 9.8 | 3.5 | 4.9 | 6.9 | 2.0 |
| BPP | 0.0 | 7.3 | 0.8 | 0.1 | 9.1 | 2.2 | 0.1 | 8.3 | 1.6 |
| IAB | 67.4 | 67.8 | 3.1 | 73.4 | 73.5 | 2.7 | 67.3 | 67.4 | 1.9 |

## 6 Concluding Remarks

The experimental evaluations presented in section 5 demonstrate the superiority of IMS over state-of-the-art model-pruning-based approaches. In particular, IMS outperforms the seminal pruning approach FP [12] across a range of challenging settings, contrasting prior observations, notably those in [8], which indicate that subsequent pruning-based approaches rarely achieve consistent improvements over FP. These results underscore the effectiveness of our proposed invertible pruning mask and selection mechanism for robust backdoor mitigation in convolutional neural networks, especially in data-scarce settings.

Comparisons with recent pruning approaches, including ANP, AWN, RNP, NFT, FMP, and BTI-DBF, emphasize the advantages of introducing mask invertibility. Our performance gains, supported by the results in Fig. 6, validate the intuition that invertible masks enable a more principled decomposition of model components based on their roles in clean versus backdoor tasks. This decomposition isolates the impact of pruning on each task, improving performance and interpretability. However, IMS's varied effectiveness against certain attack types, such as Blended and SSBA, indicates opportunities for future refinement.

When benchmarked against leading fine-tuning-based defenses such as FT-SAM and SAU, IMS delivers competitive results. While FT-SAM and SAU typically achieve stronger suppression of backdoor behavior, as reflected in their shorter and lighter ASR distribution tails, IMS consistently achieves superior RDR performance and better preserves ARR performance. This suggests that IMS excels in restoring classification accuracy post-mitigation, a capability less consistently observed in FT-SAM and SAU. This trade-off between backdoor suppression and classification recovery is especially evident in data-limited settings, underscoring the unique strengths of model pruning via selective and invertible masks. Although the performance improvements over fine-tuning are modest, achieving such results through pruning alone is notable. As fine-tuning has dominated recent research in this area, our findings provide renewed motivation for considering pruning-based alternatives.

Looking ahead, combining model pruning via selective and invertible masks with fine-tuning approaches such as FT-SAM or SAU holds promise for developing more effective backdoor defenses. By integrating the structured removal of backdoor-related components, facilitated by invertible masking, with the adaptive optimization of fine-tuning, such hybrid approaches can achieve more effective backdoor mitigation while preserving high classification accuracy. Although BTI-DBF makes an initial attempt to bridge this gap by using pruning to initialise fine-tuning, its limited success, often underperforming compared to FT-SAM, suggests that more sophisticated strategies are required. Our work lays the groundwork for such future innovations.

Finally, our preliminary evaluation of ViT-based models suggests that pruning-based approaches developed for CNNs may be transferable to transformer architectures. Future work can investigate which components of transformers are most effective targets for pruning in the context of backdoor mitigation.

## 7 Acknowledgement

Dimity Miller acknowledges ongoing support from the QUT Centre for Robotics. We acknowledge the support of QUT eResearch for providing the computing facilities required to run our experiments.

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

# A  Invertible Mask and Selection Dynamics

## A.1  Theoretical Analysis

Below, we provide a theoretical analysis demonstrating that, in the limit $\mathbf{s} \to 0$ and $k \to \infty$, the learned mask $\mathbf{a}'$ and its inverse $\bar{\mathbf{a}}'$ converge to a binary partition (i.e., they become exact complements), thereby ensuring the invertibility of the masking operation.

**Lemma 1.** *Let $\mathbf{a} \in [0,1]^N$, $\mathbf{s} \in [0,1]^N$, $k > 0$, and define the learned mask and its inverse as*

$$\mathbf{a}' = \sigma\big(k[\mathbf{a} - 0.5]\big) + \mathbf{s} \circ \sigma\big(k[(1 - \mathbf{a}) - 0.5]\big)$$
$$\bar{\mathbf{a}}' = \sigma\big(k[(1 - \mathbf{a}) - 0.5]\big) + \mathbf{s} \circ \sigma\big(k[\mathbf{a} - 0.5]\big),$$

*where $\sigma(x) = 1/(1 + e^{-x})$ denotes the sigmoid function. Then, we have $\big|\mathbf{a}' + \bar{\mathbf{a}}' - 1\big| \leq 2\mathbf{s}$. Consequently, in the limit $\mathbf{s} \to 0$, we have $\lim_{\mathbf{s} \to 0}(\mathbf{a}' + \bar{\mathbf{a}}') = 1$. Moreover, in the joint limit $\mathbf{s} \to 0$, $k \to \infty$, we obtain*

$$\mathbf{a}' \to \mathbb{1}\{\mathbf{a} > 0.5\}, \quad \bar{\mathbf{a}}' \to \mathbb{1}\{\mathbf{a} < 0.5\},$$

*where $\mathbb{1}\{\}$ denotes the indicator function. In other words, the mask $\mathbf{a}'$ becomes a hard thresholding function and the inverse mask $\bar{\mathbf{a}}'$ its exact binary complement.*

*Proof.* First, consider the case $\mathbf{s} = 0$. In this setting, the learned mask and its inverse simplify to $\mathbf{a}' = \sigma\big(k[\mathbf{a} - 0.5]\big)$ and $\bar{\mathbf{a}}' = \sigma\big(k[(1 - \mathbf{a}) - 0.5]\big) = \sigma\big(-k[\mathbf{a} - 0.5]\big) = 1 - \sigma\big(k[\mathbf{a} - 0.5]\big)$. Hence, for $\mathbf{s} = 0$, we have

$$\mathbf{a}' + \bar{\mathbf{a}}' = \sigma(k[\mathbf{a} - 0.5]) + \big[1 - \sigma(k[\mathbf{a} - 0.5])\big] = 1.$$

Now, for $\mathbf{s} > 0$, we observe that $\big|\mathbf{a}' - \sigma(k[\mathbf{a} - 0.5])\big| \leq \mathbf{s}$ and $\big|\bar{\mathbf{a}}' - \sigma(-k[\mathbf{a} - 0.5])\big| \leq \mathbf{s}$. Thus, the sum can be bounded as

$$\big|\mathbf{a}' + \bar{\mathbf{a}}' - 1\big| = \big|\big[\mathbf{a}' - \sigma(k[\mathbf{a} - 0.5])\big] + \big[\bar{\mathbf{a}}' - \sigma(-k[\mathbf{a} - 0.5])\big]\big| \leq 2\mathbf{s}.$$

Therefore, as $\mathbf{s} \to 0$, we recover the exact complementarity: $\lim_{\mathbf{s} \to 0}(\mathbf{a}' + \bar{\mathbf{a}}') = 1$. Finally, consider the limit $k \to \infty$. For any entry $a$ of the mask $\mathbf{a}$, the sigmoid function satisfies the following asymptotic behavior as $k \to \infty$:

$$\lim_{k \to \infty} \sigma\big(k[a - 0.5]\big) = \begin{cases} 1 & \text{if } a > 0.5, \\ 0 & \text{if } a \leq 0.5. \end{cases}$$

Hence, we obtain

$$\mathbf{a}' \to \mathbb{1}\{\mathbf{a} > 0.5\}, \quad \bar{\mathbf{a}}' \to \mathbb{1}\{\mathbf{a} < 0.5\}.$$

Therefore, in the joint limit $\mathbf{s} \to 0$, $k \to \infty$, the mask $\mathbf{a}'$ converges to a hard threshold at $0.5$, and $\bar{\mathbf{a}}'$ converges to its exact binary complement. This completes the proof. $\square$

## A.2  Empirical Evaluation

In Fig. 7, we examine the relationship between the mask and its inverse ($\mathbf{a}'$ and $\bar{\mathbf{a}}'$) across different values of $\mathbf{a}$ and $\mathbf{s}$ to validate the above result. Note that when an element of $\mathbf{a}'$ or $\bar{\mathbf{a}}'$ is approximately zero, it is considered pruned. These heat plots illustrate that the mask prunes the selected components ($\mathbf{s} \approx 0$) when $\mathbf{a} < 0.5$, whereas the inverse mask prunes them when $\mathbf{a} > 0.5$. In contrast, the unselected components ($\mathbf{s} \approx 1$) remain unaffected by $\mathbf{a}$ and are thus preserved by both the mask and its inverse. As discussed in Section 3, these unselected components constitute a shared backbone retained across both pruning perspectives. This analysis supports the theoretical results presented above.

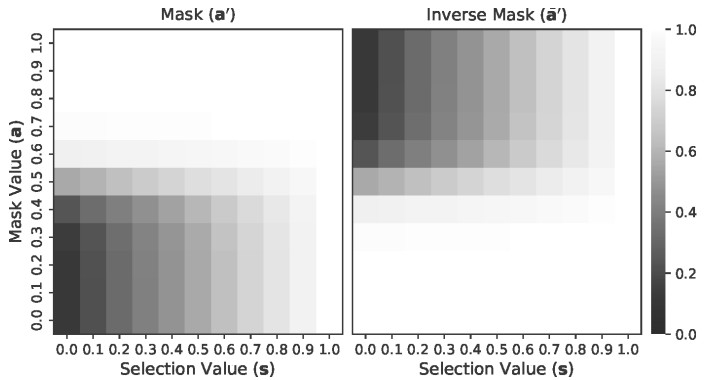

Figure 7: Values of $\mathbf{a}'$ and $\bar{\mathbf{a}}'$ as functions of $\mathbf{a}$ and $\mathbf{s}$ for $k = 20$. The boundaries $\mathbf{s} > 0.5$ and $\mathbf{s} < 0.5$ delineate components as shared and selected, respectively.

## B  Algorithms

In Algorithm 1, we summarize the training procedure of IMS.

---
**Algorithm 1** Training of IMS
---
**Require:** Dataset $\mathcal{D} = \{x_i\}_{i=1}^N$, mask parameters $\mathcal{A}$ and $\mathcal{S}$, step size $\eta$, round $R_{1,2,3}$, and bound $\epsilon$.
 1: **for** $i = 1$ to $R_1$ **do** # Mask Initialisation
 2:  Sample minibatch $(x, y) \subset \mathcal{D}$
 3:  Compute $\mathbf{g}_\mathcal{A}$ and $\mathbf{g}_\mathcal{S}$, the gradients of the objective in (4) w.r.t. $\mathcal{A}$ and $\mathcal{S}$
 4:  $\mathcal{A}, \mathcal{S} \leftarrow \text{AdamW}(\mathcal{A}, \mathcal{S}, \mathbf{g}_\mathcal{A}, \mathbf{g}_\mathcal{S}, \eta)$
 5:  $\mathcal{A} \leftarrow \text{clip}(\mathcal{A}, 0, 1)$
 6:  $\mathcal{S} \leftarrow \text{clip}(\mathcal{A}, 0, 1)$
 7: **end for**
 8: **for** $i = 1$ to $R_2$ **do** # Outer Subproblem
 9:  Sample minibatch $(x, y) \subset \mathcal{D}$
10:  $\delta \leftarrow \text{zeros\_like}(x)$
11:  **for** $i = 1$ to $R_3$ **do** # Inner Subproblem
12:    Compute $\mathbf{g}_\delta$, the gradient of the objective in (5) w.r.t. $\delta$
13:    $\delta \leftarrow \text{AdamW}(\delta, \mathbf{g}_\delta, \eta)$
14:    $\delta \leftarrow \text{clip}(\delta, -\epsilon, \epsilon)$
15:  **end for**
16:  Compute $\mathbf{g}_\mathcal{A}$ and $\mathbf{g}_\mathcal{S}$, the gradients of the objective in (6) w.r.t. $\mathcal{A}$ and $\mathcal{S}$
17:  $\mathcal{A}, \mathcal{S} \leftarrow \text{AdamW}(\mathcal{A}, \mathcal{S}, \mathbf{g}_\mathcal{A}, \mathbf{g}_\mathcal{S}, \eta)$
18:  $\mathcal{A} \leftarrow \text{clip}(\mathcal{A}, 0, 1)$
19:  $\mathcal{S} \leftarrow \text{clip}(\mathcal{A}, 0, 1)$
20: **end for**
---

## C  Performance Measures

To evaluate the effectiveness of backdoor mitigation, we adopt three performance measures: accuracy reduction ratio (ARR), attack success rate (ASR), and recovery difference ratio (RDR).

ASR assesses the efficacy of the backdoor attack by computing the accuracy on backdoor test samples, which are clean inputs modified with the trigger and labeled as the target class. Samples originally belonging to the target class are excluded from ASR computation.

ARR quantifies the impact of mitigation on clean accuracy, defined as:

$$\text{ARR} = 1 - \frac{\alpha_s}{\alpha_p}, \tag{8}$$

where $\alpha_p$ and $\alpha_s$ denote clean accuracy before and after mitigation, respectively. The normalization by $\alpha_p$ accounts for baseline accuracy differences between datasets, such as the lower baseline accuracy

typically observed in Tiny-ImageNet compared to CIFAR-10. A lower ARR (closer to zero) indicates minimal degradation in clean accuracy due to mitigation.

RDR evaluates the effectiveness of mitigation in restoring backdoor-affected samples to their correct labels, defined as:

$$\text{RDR} = 1 - \frac{\eta_s}{\alpha_p}, \tag{9}$$

where $\eta_s$ represents recovery accuracy after mitigation. Similar to ARR, RDR is normalized by the pre-mitigation clean accuracy. An ideal mitigation strategy would result in an RDR close to zero, indicating that the model effectively restores correct classification even for inputs previously manipulated by a trigger.

## D    Computational Complexity

In Fig. 8, we analyze the computational complexity of IMS across various datasets and SPC settings. The first row represents the median number of mask initialisation (blue) and outer subproblem (orange) optimization rounds required for convergence, with the error bars indicating variability based on the median absolute deviation. The second row illustrates the median time complexity of a single mask initialisation (blue), inner subproblem (green), and outer-subproblem (orange) round, plotted on the logarithmic scale. The third row shows the median total runtime of each simulation. All experiments were run on a 4-core CPU with 32GB of RAM and a H100 GPU. The key takeaway from this analysis is that IMS is scalable to larger datasets. While increased data availability generally raises computational complexity, beyond a certain point, the increase in complexity is offset by a reduction in the number of rounds needed for convergence. Notably, in the low SPC settings of CIFAR-10, small batch sizes result in substantial computational savings. Moreover, the primary computational cost is incurred during the estimation of perturbations in each step. The complexity trends for GTSRB and Tiny-ImageNet suggest that doubling the image size results in approximately a threefold increase in runtime, yet this does not hinder the applicability of IMS to ImageNet-sized datasets.

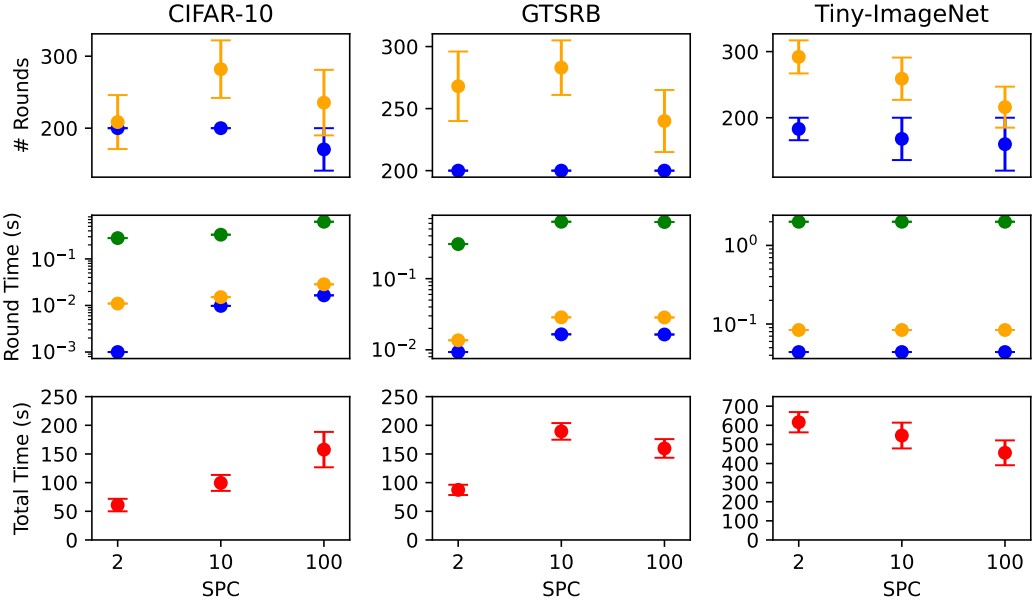

Figure 8: Median number of optimization rounds, per-round time, and total time complexity of IMS across three datasets. Error bars represent variability based on median absolute deviation. Blue: Mask Initialisation, Green: Inner Subproblem, Orange: Outer Subproblem, and Red: Total.

# E  ImageNet Subset

To assess the scalability of IMS to real-world datasets with larger image sizes, we evaluate its performance on ImageNette, a subset of ImageNet. In Table 3, we compare the median ARR, ASR, and RDR performance of IMS on ImageNette using MobileNet with those obtained on the other datasets considered in Section 5. The results indicate that IMS effectively scales to larger image tasks, as its ARR, ASR, and RDR performance on ImageNette closely aligns with that observed across the other three datasets.

Table 3: Median ARR, ASR, and RDR performance of IMS (x100) across different datasets.

| Attack | SCP | ARR | | | | ASR | | | | RDR | | | |
|---|---|---|---|---|---|---|---|---|---|---|---|---|---|
| | | CIFAR-10 | GTSRB | Tiny-ImageNet | ImageNette | CIFAR-10 | GTSRB | Tiny-ImageNet | ImageNette | CIFAR-10 | GTSRB | Tiny-ImageNet | ImageNette |
| BadNet | 1 | 24.6 | 16.4 | 37.7 | 17.9 | 16.1 | 2.5 | 14.6 | 92.4 | 40.9 | 19.4 | 48.9 | 92.3 |
| | 10 | 21.7 | 12.9 | 9.4 | 14.1 | 14.3 | 0.9 | 11.0 | 90.1 | 31.6 | 14.4 | 21.0 | 89.4 |
| | 100 | 6.4 | 7.7 | 4.2 | 7.6 | 2.4 | 0.5 | 9.5 | 63.0 | 7.8 | 8.6 | 15.3 | 60.0 |
| BPP | 1 | 27.0 | 13.9 | 12.6 | 12.6 | 2.4 | 0.4 | 0.3 | 4.7 | 28.0 | 17.7 | 15.3 | 16.1 |
| | 10 | 14.3 | 6.6 | 0.0 | 8.3 | 2.8 | 0.2 | 0.3 | 3.1 | 16.3 | 11.9 | 2.9 | 10.2 |
| | 100 | 3.2 | 4.0 | 0.0 | 2.6 | 2.0 | 0.2 | 0.4 | 1.9 | 5.8 | 9.2 | 1.4 | 4.1 |
| Inputaware | 1 | 29.5 | 17.4 | 14.6 | 13.5 | 9.7 | 0.2 | 0.7 | 2.3 | 39.7 | 24.4 | 32.3 | 19.6 |
| | 10 | 15.0 | 8.4 | 2.9 | 7.0 | 7.5 | 0.2 | 1.0 | 1.1 | 27.7 | 16.1 | 20.1 | 12.8 |
| | 100 | 4.9 | 3.7 | 0.2 | 1.7 | 9.3 | 0.1 | 1.0 | 0.9 | 18.8 | 11.1 | 20.4 | 6.1 |

# F  Mask Initialization

As part of IMS, we propose two main phases as part of our optimization framework. As mentioned in Section 4.2, our framework first initializes the mask, and then alternates between solving an inner and outer subproblem. To show the interdependence between these two phases, we compare the performance of IMS with and without mask intialization. This analysis uses the ResNet architecture, the GTSRB dataset, with a poisoning ratio of 10% and SPC of 10. The results, shown in Table 4, demonstrate that including mask initialization substantially improves ASR and RDR performance of IMS under IAB and WaNet attacks, offers modest gains in RDR performance under SSBA attack, and has a negligible effect on performance under BadNet and BPP attacks. In general, Table 4 shows that mask initialization plays a critical role in harder-to-defend attack scenarios, justifying its inclusion.

Table 4: Comparison of ASR, RDR, and ARR across different attacks when IMS is used with and without mask initialization (w/o Init).

| Attack | Defense | ASR | RDR | ARR |
|---|---|---|---|---|
| IAB | IMS w/o Init | 56.1 | 57.3 | 4.5 |
| | IMS | 1.1 | 7.1 | 4.7 |
| WaNet | IMS w/o Init | 21.8 | 21.7 | 0.0 |
| | IMS | 1.7 | 1.9 | 0.0 |
| SSBA | IMS w/o Init | 2.3 | 27.5 | 3.0 |
| | IMS | 1.0 | 26.0 | 4.0 |
| BadNet | IMS w/o Init | 0.3 | 4.0 | 3.8 |
| | IMS | 0.1 | 4.0 | 3.9 |
| BPP | IMS w/o Init | 0.0 | 7.6 | 1.8 |
| | IMS | 0.0 | 7.6 | 1.4 |

# G    Impact of $\lambda$

As part of IMS, we use $\lambda$ to weight the $\ell_1$-based sparsity term included in Equation 4 and 6. To evaluate the sensitivity of our approach to the value of $\lambda$, we have conducted experiments across a range of values using the ResNet architecture, the GTSRB dataset with a poisoning ratio of 10% and SPC of 10. The results, presented in the Table 5, show that performance remains stable for values of $\lambda$ up to 2x the default value of 10 in most cases. Exceptions being IAB and WaNet, which exhibit more sensitivity. For BadNet and BPP, performance remains robust even when $\lambda$ is increased by a factor of 10.

Table 5: Performance of IMS across varying $\lambda$ values.

| Attack | Metric | $\lambda=5$ | 10 | 15 | 20 | 25 | 50 | 75 | 100 |
|---|---|---|---|---|---|---|---|---|---|
| BadNet | ASR | 0.0 | 0.0 | 0.0 | 0.0 | 0.1 | 1.3 | 3.5 | 13.5 |
| | RDR | 3.7 | 4.4 | 4.2 | 6.0 | 4.5 | 4.7 | 6.5 | 14.1 |
| | ARR | 3.8 | 4.2 | 4.1 | 6.0 | 4.6 | 3.7 | 3.8 | 2.8 |
| BPP | ASR | 0.0 | 0.0 | 0.0 | 0.0 | 0.0 | 0.0 | 0.0 | 0.0 |
| | RDR | 7.6 | 7.6 | 7.1 | 7.4 | 7.7 | 7.6 | 6.9 | 6.4 |
| | ARR | 1.8 | 1.4 | 1.5 | 1.7 | 1.6 | 1.4 | 1.0 | 0.5 |
| IAB | ASR | 0.2 | 1.1 | 52.1 | 87.9 | 54.5 | 51.6 | 86.3 | 73.2 |
| | RDR | 5.1 | 7.1 | 54.3 | 88.5 | 55.5 | 53.2 | 86.7 | 73.2 |
| | ARR | 4.2 | 4.7 | 5.6 | 6.5 | 4.1 | 3.9 | 2.5 | 2.8 |
| SSBA | ASR | 0.7 | 1.0 | 1.0 | 2.1 | 2.7 | 60.7 | 47.8 | 85.2 |
| | RDR | 32.2 | 26.0 | 26.1 | 30.4 | 27.4 | 73.0 | 69.4 | 89.8 |
| | ARR | 4.4 | 4.0 | 4.3 | 4.3 | 2.8 | 3.1 | 2.5 | 2.7 |
| WaNet | ASR | 22.9 | 17.2 | 10.1 | 41.8 | 64.1 | 61.1 | 73.0 | 75.8 |
| | RDR | 22.1 | 16.9 | 10.5 | 41.6 | 63.9 | 61.0 | 73.0 | 75.9 |
| | ARR | 0.0 | 0.0 | 0.0 | 0.0 | 0.0 | 0.0 | 0.0 | 0.0 |

# H    Impact of $k$

As part of IMS, we use $k$ to enforce selection. To evaluate the sensitivity of IMS to the value of $k$, we conducted experiments across a range of values using the ResNet architecture, the GTSRB dataset with a poisoning ratio of 10% and SPC of 10. The results show that performance remains largely stable in the range $10 \leq k \leq 30$. When $k < 10$, binarisation of the selection parameters is less sharp, which reduces pruning specificity and degrades performance. Conversely, when $k > 30$, the steepening of the inflection point amplifies the gradient of the $\ell_1$ regularization term, discouraging selection and limiting pruning effectiveness.

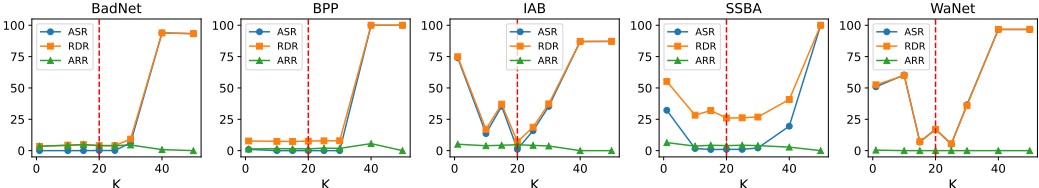

Figure 9: ARR, ASR and RDR performance of IMS for different values of $k$. Red line indicates the chosen value of 20 used in all experiments.

# I    Impact of selection

To empirically examine the trade-off between selection and ARR, we measured the relationship between the effective sparsity of $\mathcal{S}$ and ARR performance across each tested setting. We define

Table 6: Selection statistics across datasets and models. $\rho$ = Spearman's $\rho$, Min/Max = Minimum/Maximimum proportion of values $< 0.5$ or $> 0.5$ across all tested settings.

| Dataset | Model | Significant | $\rho$ | Min | Max |
|---------|-------|-------------|--------|-----|-----|
| CIFAR-10 | VGG | False | 0.053 | 1.07 | 12.52 |
| CIFAR-10 | ResNet | True | -0.164 | 1.71 | 13.03 |
| CIFAR-10 | EfficientNet | True | -0.496 | 0.38 | 2.98 |
| CIFAR-10 | MobileNet | True | -0.310 | 0.24 | 2.81 |
| GTSRB | VGG | True | 0.705 | 2.31 | 10.65 |
| GTSRB | ResNet | True | 0.384 | 2.59 | 9.85 |
| GTSRB | EfficientNet | True | -0.222 | 0.54 | 5.32 |
| GTSRB | MobileNet | True | 0.466 | 0.45 | 5.70 |
| Tiny | VGG | True | 0.680 | 2.65 | 12.95 |
| Tiny | ResNet | True | 0.577 | 2.62 | 11.14 |
| Tiny | EfficientNet | True | 0.681 | 0.24 | 5.40 |
| Tiny | MobileNet | True | 0.926 | 0.16 | 4.27 |

effective sparsity as the proportion of entries in $\mathcal{S}$ that fall below 0.5. Using Spearman's $\rho$, we assessed how sparsity relates to ARR across each dataset–model pair, and also report the observed sparsity ranges. As hypothesised, GTSRB and Tiny ImageNet show a monotonic increasing association between the proportion of selected values and ARR ($\rho > 0.4$ in most cases), which aligns with the corresponding sparsity minima and maxima. In contrast, CIFAR-10 typically exhibits a weaker, monotonic decreasing relationship. Notably, the narrow spread between the minimum and maximum values across all settings suggests that, in absolute terms, the extent of pruning remains modest relative to the total model size. While increasing $\lambda$ could theoretically drive higher sparsity, our analysis in Appendix G shows that large $\lambda$ values substantially harm ASR and RDR, while yielding only marginal gains in ARR.

## J   Backdoor Reactivation Attack

The recent work [29] introduces a method for reactivating backdoors by optimizing the adversary's trigger after a defense has been applied. In the black-box setting considered, the attacker is allowed unlimited queries to the defended model and receives the corresponding softmax outputs. Leveraging this information, the adversary computes sample-specific perturbations that, when applied to backdoor inputs, aim to restore misclassification to the original target class.

In Table 7, we report the reactivation ASR and RDR of IMS, FT-SAM, and SAU using the GTSRB dataset with 100 SPC as the evaluation setting. Overall, while IMS exhibits slightly higher susceptibility to backdoor reactivation compared to SAU, its vulnerability is comparable to FT-SAM. Notably, in 50% of the evaluated scenarios, IMS maintains a reactivation ASR below 50%.

Regarding reactivation RDR, which is the recovery accuracy of reactivation samples using their original (clean task) labels relative to the original model's accuracy, we observe only minor differences across the three defenses. This suggests that while SAU more effectively prevents targeted misclassification through backdoor reactivation, it does not necessarily mitigate untargeted misclassification. In contrast, IMS and FT-SAM exhibit similar RDR performance to SAU, with a larger proportion of reactivation samples being misclassified as the target class. Although the primary focus of this work is not on reactivation attacks, we highlight this finding as a critical consideration for future research in backdoor mitigation.

## K   Additional Results and Plots

### K.1   Model and Attack Plots

In Fig. 10 we provide the complete version of the summary results provided in the main text as Fig. 5. As discussed in section 5.2, IMS significantly improves the inconsistent ASR and RDR performance of existing pruning methods across different model architectures and attack types. However, compared

Table 7: ASR and RDR (x100) of IMS, FT-SAM, and SAU under backdoor reactivation attack on GTSRB dataset with SPC=100. Columns $\Delta_{\text{SAU}}$ and $\Delta_{\text{FT-SAM}}$ represent the performance difference of IMS compared to SAU and FT-SAM, respectively. A negative value signifies better performance for IMS (lower ASR, higher RDR).

| Attack | Model | ASR | | | | | RDR | | | | |
| | | IMS | SAU | FT-SAM | $\Delta_{\text{SAU}}$ | $\Delta_{\text{FT}}$ | IMS | SAU | FT-SAM | $\Delta_{\text{SAU}}$ | $\Delta_{\text{FT}}$ |
| --- | --- | --- | --- | --- | --- | --- | --- | --- | --- | --- | --- |
| BadNet | ResNet | 61.39 | 0.41 | 58.45 | 60.99 | 2.94 | 63.13 | 23.91 | 60.62 | 39.22 | 2.51 |
| | MobileNet | 78.18 | 44.07 | 58.01 | 34.11 | 20.17 | 79.32 | 74.64 | 60.89 | 4.69 | 18.43 |
| BPP | ResNet | 19.06 | 41.64 | 43.75 | -22.58 | -24.69 | 42.13 | 51.50 | 48.57 | -9.37 | -6.44 |
| | MobileNet | 28.34 | 26.53 | 44.14 | 1.81 | -15.81 | 49.14 | 60.13 | 52.02 | -10.99 | -2.88 |
| Inputaware | ResNet | 44.15 | 22.81 | 73.90 | 21.34 | -29.75 | 53.93 | 68.55 | 75.48 | -14.62 | -21.55 |
| | MobileNet | 37.86 | 22.90 | 51.04 | 14.96 | -13.18 | 58.85 | 70.30 | 59.10 | -11.44 | -0.24 |
| SSBA | ResNet | 98.19 | 70.68 | 84.34 | 27.51 | 13.85 | 98.31 | 95.00 | 87.53 | 3.31 | 10.78 |
| | MobileNet | 94.42 | 67.27 | 94.47 | 27.15 | -0.05 | 95.49 | 89.26 | 95.73 | 6.23 | -0.25 |
| WaNet | ResNet | 91.59 | 63.94 | 49.86 | 27.65 | 41.73 | 91.42 | 71.81 | 52.14 | 19.61 | 39.28 |
| | MobileNet | 51.03 | 43.16 | 52.83 | 7.88 | -1.80 | 57.87 | 63.33 | 55.19 | -5.46 | 2.68 |
| | | | | Mean Diff | **20.08** | **-0.66** | | | Mean Diff | **2.12** | **4.23** |

to BTI-DBF, FT-SAM, and SAU, IMS exhibits slightly higher ASR variance across the considered attacks. Interestingly, the RDR performance patterns of IMS, BTI-DBF, FT-SAM, and SAU are generally similar across attack types, with IMS showing slightly lower variance and median RDR in most cases.

## K.2 Density Plots

In Fig. 11 we present a scatter plot of the ASR vs RDR and ARR performance of each method. Note, that ideal performance is in the bottom left-hand corner of each plot. Consistent with the findings presented in Section 5.2, these plots show that IMS substantially enhances the performance of existing pruning-based methods. This is evident from the pronounced shift in result density away from the top right and bottom left corners when comparing IMS to FP, ANP, AWN, RNP, NFT, and FMP. Compared to fine-tuning works, IMS trades off some ASR performance, as indicated by greater variance along the ASR axis, for improved RDR and ARR performance. This is supported by the ARR and RDR results of BTI-DBF, FT-SAM and SAU exhibiting greater variance along these axes compared to IMS.

Figure 12 presents scatter plots of ASR versus RDR and ARR for IMS and each fine-tuning method at SPC levels of 2, 10, and 100. In line with the results discussed in Section 5.2, these plots demonstrate that IMS maintains notably more stable ARR and RDR performance under reduced data availability compared to fine-tuning methods. In particular, the ARR and RDR performance of fine-tuning methods becomes more varied compared to IMS as SPC is reduced.

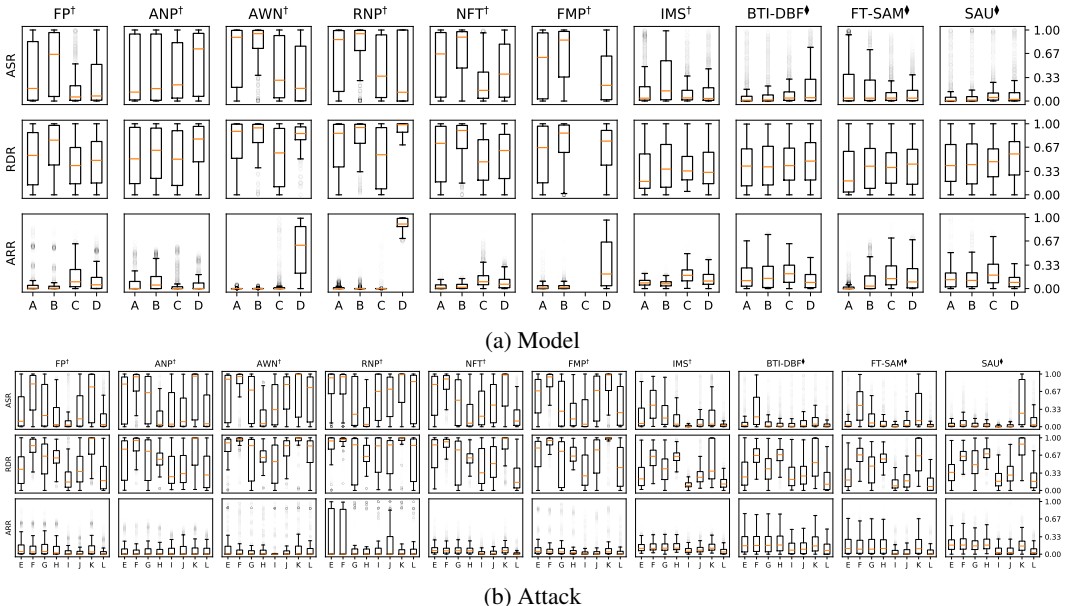

(a) Model

(b) Attack

Figure 10: Box plots illustrating ASR, RDR, and ARR results for IMS compared to all tested methods, across different Model and Attack settings. In Plot (A) A: VGG, B: ResNet, C: EfficientNet and D: MobileNet. In Plot (B) E: BadNet, F: Blended, G: LF, H: Signal, I: BPP, J: Inputaware, K: SSBA and L: WaNet.

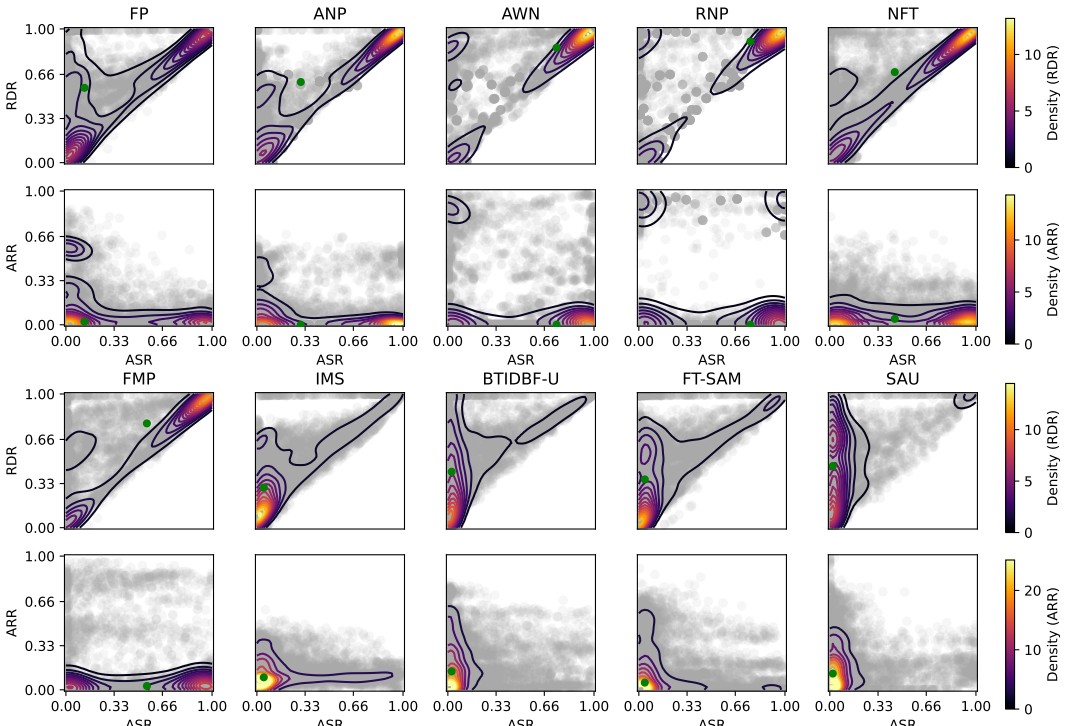

Figure 11: Scatter plots of ASR vs RDR and ASR vs RDR for IMS and various existing pruning and fine-tuning approaches. Green points denote the median values.

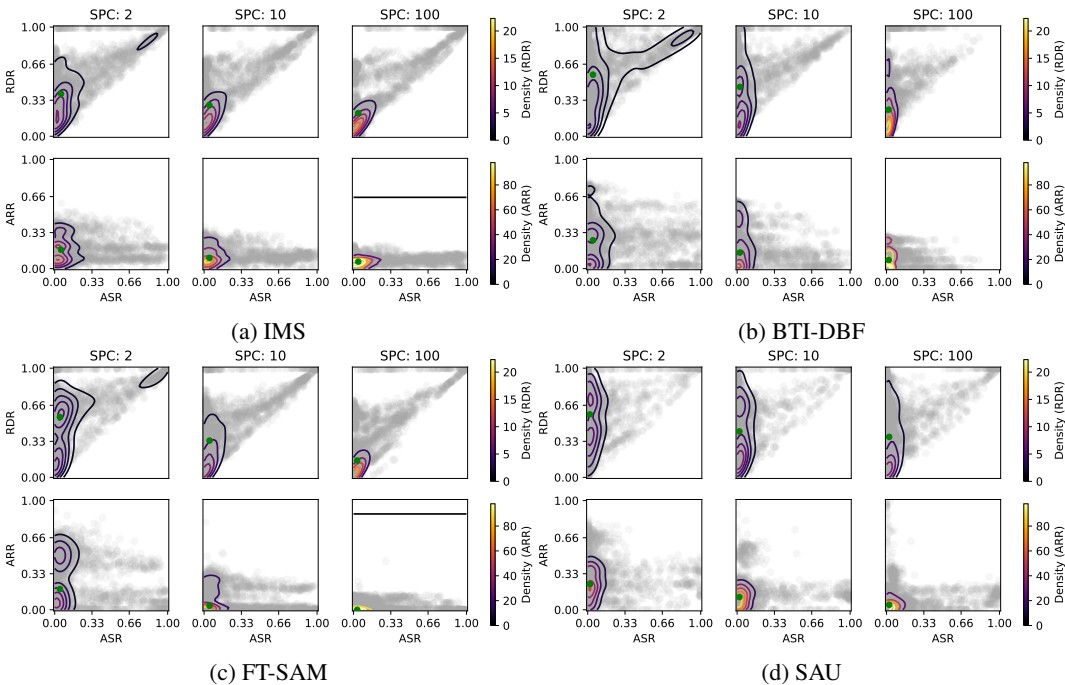

Figure 12: Scatter plots of ASR vs RDR and ASR vs RDR for IMS and various existing pruning and fine-tuning approaches, with SPC values of 2, 10, and 100. Green points denote the median values.

