# OpenReview forum: "Backdoor Mitigation via Invertible Pruning Masks"
_NeurIPS.cc/2025/Conference — NeurIPS 2025 poster_

### Official Review · Reviewer_uZpS · 2025-06-24

**Clarity:** 3
**Significance:** 3
**Originality:** 3
**Rating:** 5
**Confidence:** 4

**Summary:**

This paper proposes a novel pruning-based backdoor defense method called Invertible Masking using Selection (IMS). It introduces an invertible pruning mask and a component selection mechanism, formulated within a bi-level optimization framework. Without needing trigger samples, IMS can effectively suppress backdoor behavior while preserving clean accuracy.

**Questions:**

See weaknesses

**Ethical Concerns:**

["NO or VERY MINOR ethics concerns only"]

**Final Justification:**

The authors have addressed my all concerns

**Limitations:**

See weaknesses

**Paper Formatting Concerns:**

N.A

**Quality:**

3

**Strengths And Weaknesses:**

**Strengths**
1. The concept of invertible masks is both intuitive and effective. It provides a principled way to distinguish between clean and backdoor components, improving interpretability over existing methods that rely on one-sided heuristics.

2. The method assumes no access to trigger samples and only limited clean data, which is realistic in outsourced training scenarios. It scales well to large datasets (e.g., Tiny-ImageNet, ImageNette) with reasonable computational complexity.

3. IMS consistently achieves lower ASR and RDR than prior pruning defenses (FP, ANP, AWN, etc.), and remains competitive or superior to fine-tuning methods (FT-SAM, SAU).

**Weaknesses**
1. While the two-level training loop is central to the method, the paper lacks convergence analysis or training stability discussions.

2. The paper highlights computational savings over fine-tuning in text but provides no runtime or resource comparison (e.g., time per epoch, GPU memory). This weakens the case for pruning as a practical alternative.

3. While the proposed method is currently limited to CNNs, the paper should ideally provide at least a high-level discussion on how the key components, particularly the invertible mask and selection mechanism, might be extended to Vision Transformers (ViTs)

4. Although the proposed method achieves low ASR and RDR on average, certain attacks still result in high recovery error. The paper does not analyze these failure cases or provide examples to understand the limits of the defense. Including qualitative or case-by-case analysis of misclassified samples would improve interpretability and guide future improvements.

5. The paper introduces a selection mechanism to avoid over-pruning, but it does not clearly explain how pruning affects the model’s clean accuracy. For example, if stronger pruning removes more backdoor behavior, does it also harm clean predictions? It would be helpful to show how the method balances these two goals, perhaps by plotting how clean accuracy and backdoor removal change as pruning becomes more aggressive.

---

> ### Author Rebuttal · Authors · 2025-07-31
>
> Thank you for your thoughtful and constructive feedback. We are encouraged by the positive reception of our work and appreciate the time and effort taken to evaluate our paper. In the responses below, we address each comment individually and indicate where clarifications or revisions will be made in the paper. Please let us know if you believe the changes are inadequate or if further clarification is needed. We are eager to work collaboratively with all reviewers during the discussion period to further improve the paper.
>
> **Comment 4.1: Convergence Analysis**
>
> Unfortunately, the highly non-convex nature of the models used in our evaluation precludes any form of rigorous convergence analysis. As a middle ground, we include plots in Appendix D (Figure 8) that show the number of rounds required by the two-level training loop. These results indicate that, for mask initialization, the median number of rounds matches the early stopping threshold in half of the cases. Moreover, none of the plotted runs reach the maximum number of rounds, suggesting that early stopping occurs when the aggregate training loss plateaus. Based on your feedback, along with Comment 2.5 from Reviewer dLLr, we will add this discussion to Appendix D.
>
> **Comment 4.2: Computational Efficiency**
>
> We do not make any claim about computationally efficiency of IMS compared to existing fine-tuning methods. This is also reflected in our response to the "Experiments compute resources" section of the NeurIPS Paper Checklist. In Appendix D, we provide an evaluation of our method’s computational efficiency, reporting the number of rounds, cost per round, total run time, and the resources used to conduct these experiments.
>
> **Comment 4.3: ViTs**
>
> In response to this comment, as well as similar feedback from Reviewer k34Q, we plan to add a brief section, either in the main text or supplementary material, addressing the applicability of IMS to ViT-based architectures. Additionally, motivated by this discussion, we conducted preliminary experiments to evaluate the applicability of our method to Vision Transformers (ViTs), with promising initial results. A draft of the planned inclusion is provided below.
>
> Vision Transformers (ViTs) have gained popularity in image classification tasks in recent years. Similar to CNNs, several studies have shown that ViTs are highly overparameterized and can be pruned to improve efficiency. Notably, [1] demonstrated that ViTs can sustain high levels of pruning while maintaining performance. Building on this insight, pruning encoder blocks appears promising for extending pruning-based methods to non-CNN architectures. We hypothesize that the backdoor task may be encoded in specific embedding dimensions within each block. Therefore, applying a mask to prune these embedding dimensions is a natural first step.
>
> To assess the effectiveness of this strategy, we apply IMS to ViT-B/16 trained with the BadNet, BPP, and IAB attacks on GTSRB. The results are presented in the table below. The only modification we make to the method is the removal of the sparsity term in Equation (6), motivated by the findings of [1]. Remarkably, IMS performs very well on the BadNet and BPP attacks across all SPC settings. Although the results for IAB are less promising, this preliminary investigation demonstrates the potential of IMS and other pruning approaches for ViTs.
>
> Future work should explore which model components are most suitable for pruning, as the embedding dimension is only one option. For example, the Q, K, and V matrices offer valuable information, as demonstrated by feature selection methods such as [2] using them to improve model efficiency.
>
> [1] Xu, Kaixin, et al. "Lpvit: Low-power semi-structured pruning for vision transformers." European Conference on Computer Vision. Cham: Springer Nature Switzerland, 2024.
>
> [2] Huang, Lan, et al. "Efficient feature selection for pre-trained vision transformers." Computer Vision and Image Understanding 254 (2025): 104326.
>
> |Attack|SPC|ASR|RDR|ARR|
> |-|-|-|-|-|
> |BadNet|2|2.2|6.3|4.0|
> |BadNet|10|6.9|9.8|3.5|
> |BadNet|100|4.9|6.9|2.0|
> |BPP|2|0.0|7.3|0.8|
> |BPP|10|0.1|9.1|2.2|
> |BPP|100|0.1|8.3|1.6|
> |IAB|2|67.4|67.8|3.1|
> |IAB|10|73.4|73.5|2.7|
> |IAB|100|67.3|67.4|1.9|
>
> **Comment 4.4: Evaluation of Fail Cases**
>
> In Figure 6, we examine how ASR varies with the proportion of perturbations that cause misclassification towards the target class. This analysis shows that when this proportion is high, IMS is significantly more robust and effective. To strengthen this analysis, we will add additional plots in the appendix that show the per-attack relationship within each dataset category. However, due to NeurIPS policies regarding figure sharing, we are unable to include these plots in the rebuttal.
>
> In general, results characterized by a longer and flatter distribution along the x-axis (Prop Target Class) tend to exhibit a shorter ASR tail. This observation suggests that the variation in attack performance shown in Figure 5(b) may be explained by the difficulty of estimating the trigger distribution.
>
> **Comment 4.5: Impact of Pruning**
>
> We balance backdoor removal and clean accuracy by ensuring that the main loss terms in Equations (4) and (6), the agree and disagree components, share a common construction and therefore have comparable magnitudes. Additionally, the $\ell_{1}$ penalty in Equation (6) encourages sparsity in the selection variables, thereby discouraging unnecessary pruning.
>
> To empirically examine this trade-off, we measured the relationship between effective sparsity and ARR. We define effective sparsity as the proportion of entries in $\mathcal{S}$ and the inverse mask $\bar{\mathcal{A}}'$ that fall below 0.5. Using Spearman’s $\rho$, we assessed how these sparsity metrics relate to ARR across each dataset–model pair, and also report the observed sparsity ranges. As hypothesised, GTSRB and Tiny ImageNet show a monotonic increasing association between the proportion of selected/pruned values and ARR ($\rho > 0.5$ in most cases), which aligns with the corresponding sparsity minima and maxima. In contrast, CIFAR-10 typically exhibits a weaker, monotonic decreasing relationship. Notably, the narrow spread between the minimum and maximum values of $\bar{\mathcal{A}}'$ across all settings suggests that, in absolute terms, the extent of pruning remains modest relative to the total model size.
>
> Finally, while increasing $\lambda$ could theoretically drive higher sparsity, our analysis in response to Reviewer dLLr (Comment 2.3) shows that large $\lambda$ values substantially harm ASR and RDR, while yielding only marginal gains in ARR. We will include the full results table as well as a variant of this discussion in the supplementary material so that others can benefit from these findings.
>
> *Table caption: Significance (sig), Spearman $\rho$, and observed minimum/maximum pruning proportions (as percentages) across runs.*
>
> |Dataset|Model|Selection||| |Mask||| |
> |-------|-----|--------|--|----|----|----|--|----|----|
> |       |     |Sig     |ρ |Min |Max |Sig |ρ |Min |Max |
> |CIFAR-10|VGG|False|0.053|1.07|12.52|True|-0.321|0.15|5.63|
> |CIFAR-10|ResNet|True|-0.164|1.71|13.03|True|-0.456|0.56|6.13|
> |CIFAR-10|EfficientNet|True|-0.496|0.38|2.98|True|-0.540|0.24|2.26|
> |CIFAR-10|MobileNet|True|-0.310|0.24|2.81|False|-0.074|0.15|1.57|
> |GTSRB|VGG|True|0.705|2.31|10.65|True|0.232|0.44|4.07|
> |GTSRB|ResNet|True|0.384|2.59|9.85|False|-0.002|0.88|3.74|
> |GTSRB|EfficientNet|True|-0.222|0.54|5.32|True|-0.346|0.45|3.46|
> |GTSRB|MobileNet|True|0.466|0.45|5.70|True|0.209|0.19|2.25|
> |Tiny|VGG|True|0.680|2.65|12.95|True|0.447|1.14|5.92|
> |Tiny|ResNet|True|0.577|2.62|11.14|True|0.457|1.86|6.87|
> |Tiny|EfficientNet|True|0.681|0.24|5.40|True|0.503|0.15|3.28|
> |Tiny|MobileNet|True|0.926|0.16|4.27|True|0.845|0.06|3.43|

---

> > ### Author Response · Authors · 2025-08-05
> > **Rebuttal follow-up**
> >
> > Dear Reviewer uZpS,
> >
> > We appreciate that your time is valuable and that the review process is demanding for both authors and reviewers. However, we wanted to follow up regarding our rebuttal to see if we have adequately addressed your concerns. We went to significant effort in the short period of time available and are interested in engaging with you to improve our work.
> >
> > Thanks,
> > Authors

---

> > ### Comment · Reviewer_uZpS · 2025-08-06
> >
> > Thank you for addressing some of the issues I raised. I will update my rating, primarily due to the experiment extending the method to ViT, which demonstrates the proposed method’s applicability to all major model architectures.

---

> > > ### Author Response · Authors · 2025-08-06
> > >
> > > Dear Reviewer uZpS,
> > >
> > > Thank you for your response to our rebuttal and for considering our additional results when submitting your final score. We appreciate your suggestion to include ViT-based analysis; this addition meaningfully strengthens the paper.
> > >
> > > Thanks,
> > > Authors

---

### Official Review · Reviewer_k34Q · 2025-06-24

**Clarity:** 3
**Significance:** 2
**Originality:** 2
**Rating:** 4
**Confidence:** 4

**Summary:**

This paper proposes *Invertible Masking using Selection* (IMS) for backdoor defense. Specifically, IMS employs an invertible mask and a selection mechanism to distinguish between model components for benign and backdoor tasks. The method formulates a bi-level optimization problem to jointly learn the pruning mask and selection parameter vector derived from clean data. The authors conducted extensive experiments on various datasets, models, and attacks to validate the effectiveness of the IMS.

**Questions:**

- What is the value of $k$ in the experiments? Does it affect the defense performance?
- How does the IMS perform against the adaptive attacks? [1] This is to validate whether the IMS is still effective when the benign and backdoor components are less separated.
- What if there are multiple backdoor targets, such as all-to-all attacks?
- Other questions are listed in the weaknesses section.


[1] Revisiting the assumption of latent separability for backdoor defenses

**Ethical Concerns:**

["NO or VERY MINOR ethics concerns only"]

**Final Justification:**

As most of my concerns are addressed, I decided to raise my score to 4.

Overall, this paper demonstrates strong defense performance and well-designed experiments. My primary concern was the novelty of the methodology. While some concerns remain, the author's rebuttal has adequately addressed my main issues, such as the generalizability of IMS and the motivation behind the approach. Therefore, I have decided to raise my score to 4.

**Limitations:**

While the author has claimed that extending the IMS to ViT-based models requires significant efforts, it is a critical limitation given the fact that the backdoor defense against small (ConvNets) models becomes saturated. In contrast, the fine-tuning approaches are more general.

**Paper Formatting Concerns:**

No.

**Quality:**

3

**Strengths And Weaknesses:**

## Strengths

- Overall, the paper is well-written and easy to follow, with clear experimental settings and method formulations.
- IMS is evaluated across a range of backdoor attacks, datasets, and model architectures. The authors also consider various advanced defense methods, which increase the soundness of the paper.
- The IMS achieves a promising defense performance when clean data is limited.

## Weaknesses

- The motivation behind *invertible-masks*, i.e., the mask and its inverse, is unclear. For example, in ANP, they use bi-level optimization to ensure both benign performance (a larger mask will result in benign performance degradation) and backdoor mitigation (a smaller inverse mask will lead to ineffective mitigation), which implicitly uses *invertible-masks* in their defense. Thus, why does IMS's design result in improved defense performance compared to ANP/RNP? What if you use the ANP to prune all model parameters, not just the channel? This will validate whether the improvement stems from increasing the mask sizes or the design of the explicit *invertible-masks*.
- The defense performance appears to be inferior to several existing methods for ASR and ARR (when SPC is greater than 10), such as FT-SAM and SAU (see Fig. 4 in the submission). That being said, can you elaborate on the advantages of IMS over these methods, in which clean data is not as limited? While I acknowledge that practical defenders may access limited clean data, evaluating against the scenario where SPC is 2 or even fewer is not necessary, since most defenses, even the IMS, are not effective in this case (a significant sacrifice in clean accuracy).

---

> ### Author Rebuttal · Authors · 2025-07-31
>
> Thank you for your thoughtful and constructive feedback. We are encouraged by the positive reception of our work and appreciate the time and effort taken to evaluate our paper. In the responses below, we address each comment individually and indicate where clarifications or revisions will be made in the paper. Please let us know if you believe the changes are inadequate or if further clarification is needed. We are eager to work collaboratively with all reviewers during the discussion period to further improve the paper.
>
> **Comment 3.1: Novelty Compared to ANP and RNP**
>
> While ANP employs a bi-level optimization strategy, it only considers the subtractive effect of pruning, i.e., how model behavior changes when components are removed. Specifically, in ANP, the maximization step uses $\mathbf{m} + \mathbf{\delta}$ (Equation 8 in their paper), where $\mathbf{m}$ is a learnable binary mask and $\mathbf{\delta}$ is the perturbation. However, the outer minimization step operates solely on $\mathbf{m}$, and there is no mechanism to explicitly evaluate how pruned components behave when retained. Thus, although $\mathbf{\delta}$ can transiently activate masked channels, it does not constitute a true inverse mask. Components retained by $\mathbf{m}$ are never pruned, and components pruned by $\mathbf{m}$ are not systematically evaluated in isolation.
>
> RNP also adopts a subtractive approach. During the \emph{recovering} stage, a mask is learned to restore benign accuracy after unlearning, and this same mask is applied to the original model parameters. At no point does RNP evaluate the behavior of components under an inverse pruning scheme.
>
> In contrast, IMS explicitly introduces an invertible masking mechanism that enables us to analyze how pruned components behave when retained and how retained components behave when excluded. This dual perspective (enabled by structured, complementary masks) allows for a more comprehensive assessment of component behavior and leads to more targeted and reliable pruning.
>
> Regarding your question about where pruning is applied: all three methods (ANP, RNP, and IMS) target the same set of model components (e.g., convolutional channels), hence the overall pruning scope is consistent. Therefore, our observed performance improvements cannot be attributed to differences in pruning granularity or scope. Rather, they stem from IMS’s unique selection strategy and the dual evaluation framework enabled by invertible masks.
>
> Our core motivation, as discussed in Section 4.1, is to address a fundamental limitation in prior pruning-based defenses: the assumption that the backdoor can be removed  purely through subtraction (i.e., pruning suspected backdoor components). By evaluating both pruning and retention effects through complementary masks, IMS establishes a more robust criterion for component selection, particularly in contrast to ANP and RNP. We will revise Section 4.1 to clarify these distinctions and highlight the novelty and impact of our approach.
>
> **Comment 3.2: Fine-tuning Comparison**
>
> To clarify the performance differences, Table 3 shows that in low-data settings (2 and 10 SPC), IMS achieves significantly higher RDR and ARR compared to BTI-DBF, FT-SAM, and SAU. Moreover, as shown in Figure 4, IMS exhibits lower variance in ARR across all SPC levels, suggesting improved consistency. While FT-SAM outperforms IMS in terms of median ASR and ARR at 100 SPC, IMS still achieves better RDR than BTI-DBF and SAU, with comparable ARR. This highlights its competitiveness even in higher-data regimes, although its primary advantage over state-of-the-art approaches emerges when data availability is limited, a scenario where fine-tuning methods are often less effective or even infeasible. Nonetheless, we agree that SPC values below 2 are rarely practical. Our evaluation at 2 SPC is intended to benchmark the lower limit of applicability across all approaches, and we acknowledge that performance degradation is expected in such extreme cases.
>
> That said, IMS offers additional advantages beyond performance, particularly in higher-SPC settings. As noted in the Introduction, Conclusion, and by Reviewers uZpS and dLLr, pruning-based approaches (such as IMS) offer improved interpretability compared to fine-tuning. Specifically, IMS yields a learned mask that explicitly identifies components associated with the backdoor task, offering greater transparency into the mitigation process. In contrast, fine-tuning approaches operate through implicit gradient-based updates, making it more difficult to attribute behavioral changes to specific model components. We will revise Section 5.2 to more clearly emphasize this interpretability benefit.
>
> **Comment 3.3: Importance of K**
>
> In all experiments, we set $k = 20$. In response to this comment and related ones from other reviewers, we have conducted an ablation study to evaluate the sensitivity of our approach to the value of $k$. The table below presents results using a representative subset of our experiments using the ResNet model and the GTSRB dataset with a poisoning ratio of 10\% and SPC of 10.
>
> The results show that performance remains largely stable in the range $10 < k < 30$. When $k < 10$, the binarisation of the selection parameters is less sharp, which reduces pruning specificity and degrades performance. Conversely, when $k > 30$, the steepening of the inflection point amplifies the gradient of the $\ell_1$ regularization term, discouraging selection and limiting pruning effectiveness.
>
> We will include these results, along with additional ablation studies on the effect of $\lambda$ and the importance of mask initialization, in the final paper.
>
> |Attack|k|ASR|RDR|ARR|
> |------|--|----|----|----|
> |BadNet|1|0.0|3.4|3.6|
> |BadNet|10|0.0|4.4|4.2|
> |BadNet|15|0.0|4.8|4.8|
> |BadNet|20|0.1|4.0|4.0|
> |BadNet|25|0.1|4.0|4.0|
> |BadNet|30|6.2|9.2|4.5|
> |BadNet|40|93.9|93.9|0.8|
> |BadNet|50|93.3|93.3|0.0|
> |BPP|1|1.0|7.7|1.3|
> |BPP|10|0.0|7.4|1.6|
> |BPP|15|0.0|7.3|1.5|
> |BPP|20|0.0|7.6|1.4|
> |BPP|25|0.0|7.8|2.0|
> |BPP|30|0.0|7.9|1.8|
> |BPP|40|100.0|100.0|5.6|
> |BPP|50|100.0|100.0|0.0|
> |IAB|1|74.1|74.9|5.1|
> |IAB|10|13.6|17.0|3.9|
> |IAB|15|35.4|37.1|4.3|
> |IAB|20|1.1|7.1|4.7|
> |IAB|25|15.9|18.7|4.2|
> |IAB|30|35.3|37.3|3.7|
> |IAB|40|87.0|87.1|0.0|
> |IAB|50|87.1|87.2|0.0|
> |SSBA|1|32.2|55.2|6.4|
> |SSBA|10|1.7|28.3|3.6|
> |SSBA|15|0.9|32.0|4.3|
> |SSBA|20|1.0|26.0|4.0|
> |SSBA|25|1.0|26.2|4.3|
> |SSBA|30|2.1|26.8|4.0|
> |SSBA|40|19.5|40.8|2.8|
> |SSBA|50|100.0|100.0|0.0|
> |WaNet|1|50.9|52.5|0.4|
> |WaNet|10|60.4|59.9|0.0|
> |WaNet|15|6.8|7.2|0.0|
> |WaNet|20|17.2|16.9|0.0|
> |WaNet|25|5.1|5.5|0.0|
> |WaNet|30|36.6|35.9|0.0|
> |WaNet|40|96.6|96.8|0.0|
> |WaNet|50|96.6|96.8|0.0|
>
> **Comment 3.4: Considered Attacks**
>
> Our current study includes several adaptive backdoor attacks such as BPP, IAB, SSBA, and WaNet that dynamically modify their trigger based on the input. As shown in Figure 5(b), IMS demonstrates comparable performance across both adaptive and static attacks (e.g., BadNet, Blended, LF, and Signal), suggesting robustness even when backdoor patterns are input-dependent.
>
> While we acknowledge the value of evaluating IMS against a broader set of attacks, including the one you mentioned, as well as multi-trigger, multi-target, and all-to-all backdoor settings, such experiments are currently beyond our computational budget. Our full set of experiments, including additional results omitted from the main paper due to space constraints, already exceeds 25 TB in data storage. Nonetheless, we view this as an important direction for future work.
>
> Importantly, our evaluation protocol is aligned with current best practices in the literature, including those used in SAU, FT-SAM, and the BackdoorBench benchmarking paper. Although the mentioned attack appears to reduce latent space separability, the visualizations provided in that work focus on feature distributions from the final dense layer. In contrast, IMS selects and prunes components in all convolutional layers, making the comparison less direct.
>
> **Comment 3.5: ViTs**
>
> To address the important point of evaluating IMS on ViT-based architectures, which was also raised by Reviewer k34Q (Comment 4.3), we will include a brief discussion in the main text (space permitting, otherwise in the appendix) regarding the applicability and potential extensions of IMS to ViT-based architectures.
>
> Motivated by this feedback, we have conducted a preliminary set of experiments applying a modified version of IMS to ViT models backdoored by BadNet, BPP, and IAB. The results are highly promising: IMS effectively mitigates both BadNet and BPP attacks across all three considered SPC settings.  While our ViT adaptation is still at an exploratory stage and does not yet include all architectural adjustments required for full generalization, these initial findings suggest that the core ideas of IMS can extend beyond ConvNets. For a more detailed description of the modified approach and corresponding results, please refer to our response to Comment 4.3.

---

> > ### Author Response · Authors · 2025-08-05
> > **Rebuttal follow-up**
> >
> > Dear Reviewer k34Q,
> >
> > We appreciate that your time is valuable and that the review process is demanding for both authors and reviewers. However, we wanted to follow up regarding our rebuttal to see if we have adequately addressed your concerns. We went to significant effort in the short period of time available and are interested in engaging with you to improve our work.
> >
> > Thanks,
> > Authors

---

> > > ### Comment · Reviewer_k34Q · 2025-08-07
> > >
> > > Thank you for addressing my concerns. I have adjusted my score accordingly.

---

### Official Review · Reviewer_dLLr · 2025-07-01

**Clarity:** 3
**Significance:** 2
**Originality:** 2
**Rating:** 4
**Confidence:** 4

**Summary:**

This paper introduces a novel model pruning-based strategy for defending against backdoor attacks in deep learning. While model pruning is recognized for its interpretability and robustness—especially valuable in low-data scenarios—existing methods struggle to precisely identify and remove parameters responsible for malicious backdoor behaviors. In contrast, fine-tuning-based defenses have gained popularity due to their effectiveness, but they often lack the same level of interpretability. The authors propose an innovative approach that incorporates two key components: (1) a learned selection mechanism to pinpoint parameters critical to both the main task and the backdoor behavior, and (2) an invertible pruning mask, which serves dual purposes—removing the backdoor while preserving it through its inverse for analysis. Through extensive experiments, the method demonstrates superior performance compared to existing pruning-based defenses and achieves results competitive with state-of-the-art fine-tuning approaches.

**Questions:**

1. **Missing Ablation Studies**
   The proposed method comprises multiple components, including initialization, selection, masking, and perturbation generation. However, the individual contribution and effectiveness of each component are not adequately analyzed through ablation studies.

2. **Unclear Choice of \$\lambda\$**
   The paper mentions that \$\lambda\$ is initially set to 0 and later increased to 10. However, the rationale behind this choice, the schedule for its adjustment, and its impact on the final performance remain unclear and underexplained.

3. **Scale of \$\epsilon\$**
   In adversarial perturbation-based methods, the magnitude of the perturbation \$\epsilon\$ is a critical factor. If the trigger exceeds this perturbation scale, its effectiveness cannot be ensured. In this work, \$\epsilon\$ is set to 1, which allows unrestricted modifications to the image. Based on prior experience, such a large value may lead to ill-posed perturbations and result in unrealistic or uninformative adversarial examples.

4. **Unexplored Parameter Choices**
   Important hyperparameters, such as the number of rounds \$R\_1\$, \$R\_2\$, and \$R\_3\$, are introduced but not analyzed or justified. Their influence on performance should be investigated.

**Ethical Concerns:**

["NO or VERY MINOR ethics concerns only"]

**Final Justification:**

Although the novelty of each component or the effectiveness compared to a fine-tuning-based method in all cases may be challenged from some perspectives, I think 1) combining existing ideas/intuitions to achieve a new SOTA is a worth-recognized contribution. 2) Beating all existing methods is not the necessary condition for a good paper, as long as it makes a contribution in this direction.

So, I would like to raise my score to support more research on pruning-based backdoor defense.

**Limitations:**

Yes

**Paper Formatting Concerns:**

Typos:

Line 249: ANW - > AWN

**Quality:**

3

**Strengths And Weaknesses:**

### **Strengths**

1. **Effective Backdoor Mitigation**
   The proposed method outperforms existing pruning-based defenses and achieves performance comparable to state-of-the-art fine-tuning approaches. This makes it a competitive and practical solution for mitigating backdoor behaviors.

2. **Innovative Use of Invertible Pruning Mask**
   The introduction of an *invertible pruning mask* is a novel contribution. It enables dual functionality—removing backdoor effects while preserving them in an inverse form. This not only enhances defense but also provides interpretability and facilitates analysis of backdoor mechanisms.

3. **Comprehensive and Rigorous Experiments**
   The paper presents a thorough experimental evaluation across diverse models, datasets, attack types, and poisoning ratios. Notably, the authors run repeated trials and report statistical results using box plots—an important but often overlooked practice in this domain.

---

### **Weaknesses**

1. **Suboptimal Performance Compared to Fine-Tuning**
   While the method significantly improves over existing pruning-based defenses, its performance still lags behind the best fine-tuning-based approaches (e.g., Figure 4 and Table 3). Nonetheless, I agree with the authors that innovation in pruning remains an important and valuable research direction.

2. **Limited Novelty in Certain Components**
   Some elements of the method—such as the use of adversarial noise as a trigger surrogate (similar to SAU), the concepts of clean/backdoor neurons (seen in ANP and RNP), and learnable masks (explored in AWN and ANP)—have been discussed in prior work. These similarities somewhat reduce the novelty. However, the proposed selection process, particularly the insight that many neurons are shared between clean and backdoor behaviors, is both original and impactful.

---

### **Overall Evaluation**

This paper makes a solid and meaningful contribution to pruning-based backdoor defense. Although fine-tuning methods currently lead in performance, the proposed approach advances the field with a well-motivated design and rigorous experimentation. Continued innovation in pruning is essential and deserves recognition.

---

> ### Author Rebuttal · Authors · 2025-07-31
>
> Thank you for your valuable feedback. In our responses below, we address each comment individually and indicate where clarifications or revisions will be made in the paper. Please let us know if you feel the changes are inadequate or if further clarification is needed. We are eager to work collaboratively with all reviewers during the discussion period to further improve the paper.
>
> **Comment 2.1: Performance Relative to Fine-Tuning**
>
> As shown in Table 3, our approach, IMS, achieves significantly better RDR and ARR performance compared to state-of-the-art fine-tuning approaches, BTI-DBF, FT-SAM, and SAU, in low-data settings (2 and 10 SPC), which we identify as a key target scenario for pruning-based defenses. Moreover, Figure 4 also shows that IMS has reduced ARR variance compared to each fine-tuning baseline in all SPC settings. While FT-SAM outperforms IMS at 100 SPC, IMS still demonstrates superior RDR performance compared to BTI-DBF and SAU, with comparable ARR. These results support our central claim that our approach offers superior mitigation performance under limited data availability, where fine-tuning approaches are often less effective or impractical.
>
> **Comment 2.2: Novelty**
>
> We agree that our proposed selection process is a a key innovation that distinguishes IMS from existing pruning-based defenses. We would also like to highlight other aspects that contribute to the broader novelty of our approach.
>
> 1) While both IMS and SAU employ bi-level optimization, their goals differ fundamentally. SAU applies bi-level optimization to fine-tune the model, whereas IMS uses it to guide model component selection for pruning. As a result, the inner and outer subproblems in IMS are fundamentally distinct in design and purpose from those in SAU.
>
> 2) In contrast to ANP and RNP, which only assess the subtractive effect of pruning (i.e., how model behavior changes when components are removed), IMS introduces an invertible masking mechanism that also evaluates model behavior when the roles of pruned and retained model components are reversed (i.e., under the inverse mask). This more comprehensive dual perspective enables more precise and reliable identification, and subsequent pruning, of backdoor-related components.
>
> 3) Our adversarial objective is explicitly formulated for a pruning setting, rather than fine-tuning, and interacts with our invertible masking mechanism in a principled way to guide component selection.
>
> We will revise the paper to more clearly highlight and elaborate on the core methodological contributions of IMS.
>
> **Comment 2.3: Missing Ablation Studies**
>
> To address this comment, as well as related ones made by other reviewers, we have conducted several additional ablation experiments that will be included in the paper. First, to assess the contribution of each optimization phase, we evaluated the impact of the mask initialization step on overall performance. As discussed in our response to Reviewer gsLE (Comment 1.1), this step encourages pruning of the inverse mask, thereby forcing the inner subproblem to rely on backdoor-related components. The results, provided with Comment 1.1, show that this step significantly improves performance in more challenging attack scenarios (e.g., IAB and WaNet), supporting its inclusion.
>
> Second, to assess the impact of $k$ in the proposed selection mechanism, we have evaluated the performance of IMS across a range of $k$ values. The results show that stronger binarisation of the selection parameters, via a larger $k$, leads to improved performance up to a certain point, beyond which gains plateau or diminis. This observation aligns with our original claim in Section 4.1. We present detailed results and further discussion in our response to Reviewer k34Q (Comment 3.3).
>
> Regarding $\lambda$, we employ a simple single-step schedule, i.e., we initially set it to $0$ and increases it to $10$ halfway through the $R_2$ rounds. This allows the mask and channel selection parameters to be updated without $\ell_1$-norm regularization in the early stages, promoting a greedy solution that is subsequently refined once regularization is introduced. To evaluate the sensitivity of our approach to the value of $\lambda$, we have conducted experiments across a range of values using the ResNet architecture and the GTSRB dataset with a poisoning ratio of 10\% and SPC of 10. The results, presented in the table below, show that performance remains stable for values of $\lambda$ up to 2x the default in most cases, except for IAB and WaNet, which exhibit more sensitivity. For BadNet and BPP, performance remains robust even when $\lambda$ is increased by a factor of 10. Finally, we emphasize that the same values of $\lambda$, $k$, and $R_{1,2,3}$ are used consistently across all our experiments.
>
> | Attack | λ   | ASR  | RDR  | ARR  |
> |--------|-----|------|------|------|
> | BadNet |   5 |  0.0 |  3.7 |  3.8 |
> | BadNet |  10 |  0.0 |  4.4 |  4.2 |
> | BadNet |  15 |  0.0 |  4.2 |  4.1 |
> | BadNet |  20 |  0.0 |  6.0 |  6.0 |
> | BadNet |  25 |  0.1 |  4.5 |  4.6 |
> | BadNet |  50 |  1.3 |  4.7 |  3.7 |
> | BadNet |  75 |  3.5 |  6.5 |  3.8 |
> | BadNet | 100 | 13.5 | 14.1 |  2.8 |
> | BPP    |   5 |  0.0 |  7.6 |  1.8 |
> | BPP    |  10 |  0.0 |  7.6 |  1.4 |
> | BPP    |  15 |  0.0 |  7.1 |  1.5 |
> | BPP    |  20 |  0.0 |  7.4 |  1.7 |
> | BPP    |  25 |  0.0 |  7.7 |  1.6 |
> | BPP    |  50 |  0.0 |  7.6 |  1.4 |
> | BPP    |  75 |  0.0 |  6.9 |  1.0 |
> | BPP    | 100 |  0.0 |  6.4 |  0.5 |
> | IAB    |   5 |  0.2 |  5.1 |  4.2 |
> | IAB    |  10 |  1.1 |  7.1 |  4.7 |
> | IAB    |  15 | 52.1 | 54.3 |  5.6 |
> | IAB    |  20 | 87.9 | 88.5 |  6.5 |
> | IAB    |  25 | 54.5 | 55.5 |  4.1 |
> | IAB    |  50 | 51.6 | 53.2 |  3.9 |
> | IAB    |  75 | 86.3 | 86.7 |  2.5 |
> | IAB    | 100 | 73.2 | 73.2 |  2.8 |
> | SSBA   |   5 |  0.7 | 32.2 |  4.4 |
> | SSBA   |  10 |  1.0 | 26.0 |  4.0 |
> | SSBA   |  15 |  1.0 | 26.1 |  4.3 |
> | SSBA   |  20 |  2.1 | 30.4 |  4.3 |
> | SSBA   |  25 |  2.7 | 27.4 |  2.8 |
> | SSBA   |  50 | 60.7 | 73.0 |  3.1 |
> | SSBA   |  75 | 47.8 | 69.4 |  2.5 |
> | SSBA   | 100 | 85.2 | 89.8 |  2.7 |
> | WaNet  |   5 | 22.9 | 22.1 |  0.0 |
> | WaNet  |  10 | 17.2 | 16.9 |  0.0 |
> | WaNet  |  15 | 10.1 | 10.5 |  0.0 |
> | WaNet  |  20 | 41.8 | 41.6 |  0.0 |
> | WaNet  |  25 | 64.1 | 63.9 |  0.0 |
> | WaNet  |  50 | 61.1 | 61.0 |  0.0 |
> | WaNet  |  75 | 73.0 | 73.0 |  0.0 |
> | WaNet  | 100 | 75.8 | 75.9 |  0.0 |
>
> **Comment 2.4: Trigger Bound**
>
> In our approach, we estimate the perturbation $\delta$ using full gradients with the AdamW optimizer (rather than using only gradient signs), and project the updates onto an $\ell\_{\infty}$-ball at each step. Specifically, we set $\epsilon = 1$ as a hard upper bound, not a target magnitude, on $\|\delta\|\_{\infty}$. Due to the use of AdamW optimizer, which incorporates decoupled weight decay, $\|\delta\|\_{2}$  is implicitly regularized and shrinks over time. As a result, the actual perturbation magnitudes remain well below the $\epsilon$ bound in practice. This behavior stands in contrast to sign-based methods (e.g., FGSM or PGD variants used in SAU), where large $\epsilon$ values often push the solution toward the boundary, directly determining the perturbation size.
>
> To clarify this distinction, we will revise Section 4.2 to explicitly explain the role of $\epsilon$ in our inner subproblem formulation.
>
> **Comment 2.5: Hyperparameters**
>
> As mentioned in our response to Comment 2.3, we have conducted additional ablation studies to evaluate the sensitivity of IMS to $\lambda$. We have also explored the impact of $k$ on component selection (Reviewer k34Q, Comment 3.3), and the importance of each optimization phase, particularly mask initialization (Reviewer gsLE, Comment 1.1). We hope these new experiments address your concerns regarding the setting of the values of the hyperparameters. Please note that all hyperparameters were held fixed across all 864 experimental settings involving different combinations of attacks, model architectures, datasets, poisoning ratios, and SPC values.
>
> We acknowledge that the number of rounds used for each phase was not discussed in the paper. Specifically, $R_1$ and $R_2$ (both set to 200) serve as upper bounds on the number of iterations for mask initialization and the outer subproblem in Algorithm 1, respectively. As shown in Figure 8 of the supplementary material, the median and median absolute deviation of the actual iteration counts indicate that, for mask initialization, only half the plotted cases reach the maximum threshold, while for the outer subproblem, none do. We will add a clarifying note on this in Section D of the supplementary material.

---

> > ### Author Response · Authors · 2025-08-05
> > **Rebuttal follow-up**
> >
> > Dear Reviewer dLLr,
> >
> > We appreciate that your time is valuable and that the review process is demanding for both authors and reviewers. However, we wanted to follow up regarding our rebuttal to see if we have adequately addressed your concerns. We went to significant effort in the short period of time available and are interested in engaging with you to improve our work.
> >
> > Thanks,
> > Authors

---

> ### Comment · Reviewer_dLLr · 2025-08-05
>
> After reading the rebuttals, I don't have many concerns.
>
> Although the novelty of each component or the effectiveness compared to a fine-tuning-based method in all cases may be challenged from some perspectives, I think 1) combining existing ideas/intuitions to achieve a new SOTA is a worth-recognized contribution. 2) Beating all existing methods is not the necessary condition for a good paper, as long as it makes a contribution in this direction.
>
> So, I would like to raise my score to support more research on pruning-based backdoor defense.
>
> BTW, I hope the authors can release their code to enhance the reproducibility of this area.

---

> > ### Author Response · Authors · 2025-08-05
> >
> > Dear Reviewer dLLr,
> >
> > We appreciate your response to our rebuttal and for supporting pruning-based backdoor defences. If accepted, we will make our code (provided in the supplementary material) publicly available to ensure reproducibility.
> >
> > Thanks,
> > Authors

---

### Official Review · Reviewer_gsLE · 2025-07-03

**Clarity:** 3
**Significance:** 3
**Originality:** 3
**Rating:** 5
**Confidence:** 4

**Summary:**

This paper presents an improved pruning-based backdoor defense by introducing a dual-optimization framework that simultaneously searches for reverse-engineered triggers and identifies backdoor-related neurons in the model. The method leverages the behavioral differences between trigger and clean inputs to more accurately locate neurons associated with the backdoor. The authors conduct experimental evaluations and demonstrate the effectiveness of their design.

**Questions:**

1. Could the authors provide detailed ablation experiments for each loss term in Equation 7?
2. There appears to be an inconsistency in notation in Equation 7 and line 224, where \tilde{p}_{\mathcal{A}'} should likely be \hat{p}_{\mathcal{A}'} based on the context. Could the authors clarify and correct this notation?
3. Could the authors provide clearer visualizations for Figures 3 and 5?

**Ethical Concerns:**

["NO or VERY MINOR ethics concerns only"]

**Final Justification:**

The authors have addressed my main concerns, and I maintain my positive recommendation.

**Limitations:**

yes

**Paper Formatting Concerns:**

/

**Quality:**

4

**Strengths And Weaknesses:**

Strengths
1. The paper introduces an innovative approach to pruning-based backdoor defense by implementing a three-way classification of neurons (backdoor-related, clean-related, and unselected) and incorporating an invertible mask mechanism.

2. The experimental results demonstrate that the proposed method can effectively remove backdoors compared to  existing pruning-based defenses, while preserving the clean accuracy.

3. The paper provides thorough experimental validation across a wide range of attack methods and compares against multiple defensive approaches.

4. The paper is well-written.

Weaknesses

1. The method incorporates multiple loss terms (particularly in Equation 7), but lacks ablation studies to demonstrate the individual contribution and necessity of each term. The authors should consider adding more ablation studies.

2. The data visualization in key figures (e.g., Figures 3 and 5) suffers from small graph sizes, making it challenging to accurately interpret the results.

---

> ### Author Rebuttal · Authors · 2025-07-29
>
> Thank you for your thoughtful and constructive feedback. We are encouraged by the positive reception of our work and appreciate the time and effort taken to evaluate our paper. In the responses below, we address each comment individually and indicate where clarifications or revisions will be made in the paper. Please let us know if you believe the changes are inadequate or if further clarification is needed. We are eager to work collaboratively with all reviewers during the discussion period to further improve the paper.
>
> **Comment 1.1: Additional ablation studies**
>
> While our method integrates multiple loss terms, we note that isolating the effect of each individual term is not straightforward due to their interdependence across the optimization phases (i.e., mask initialization, inner subproblem, and outer subproblem). Instead, as a practical compromise, we have conducted a phase-level ablation to assess the contribution of key components.
>
> Specifically, since the goal of mask initialization is to encourage the inverse mask to prune clean-task-related components, thereby pushing the inner subproblem to rely more heavily on backdoor-related components, we evaluated the impact of omitting this step. We show the results in the below Table for a subset of scenarios using the ResNet architecture and GTSRB dataset, with a poisoning ratio of 10\% and SPC of 10. The results demonstrate that including mask initialization substantially improves ASR and RDR performance of IMS under IAB and WaNet attacks, offers modest gains in RDR performance under SSBA attach, and has negligible effect on performance under BadNet and BPP attacks. These findings suggest that mask initialization plays a critical role in harder-to-defend attack scenarios, justifying its inclusion in our framework.
>
> |Attack |Defense       |ASR |RDR |ARR |
> |-------|--------------|----|----|----|
> |IAB    |IMS w/o Init  |56.1|57.3|4.5 |
> |IAB    |IMS           |1.1 |7.1 |4.7 |
> |WaNet  |IMS w/o Init  |21.8|21.7|0.0 |
> |WaNet  |IMS           |1.7 |1.9 |0.0 |
> |SSBA   |IMS w/o Init  |2.3 |27.5|3.0 |
> |SSBA   |IMS           |1.0 |26.0|4.0 |
> |BadNet |IMS w/o Init  |0.3 |4.0 |3.8 |
> |BadNet |IMS           |0.1 |4.0 |3.9 |
> |BPP    |IMS w/o Init  |0.0 |7.6 |1.8 |
> |BPP    |IMS           |0.0 |7.6 |1.4 |
>
>
> **Comment: 1.2** Figure Readability
>
> We will include all figures in vector-based PDF format to support proper scaling and legibility for online readers. We will also increase the font size of figure text where appropriate to improve readability in print.
>
> **Comment: 1.3** Notation Error
>
> We appreciate your attention to detail. We will correct the notation from $\tilde{p}\_{\mathcal{A}'}$ to $\hat{p}\_{\mathcal{A}'}$ in the mentioned equation and corresponding text.

---

> > ### Comment · Reviewer_gsLE · 2025-08-05
> >
> > The authors have addressed my main concerns.

---

### Decision · Program_Chairs · 2025-09-17

**Decision:**

Accept (poster)

**Comment:**

This paper proposes a new method to detect backdoor-affected neurons using a dual-optimization framework that simultaneously selects neurons and sample-dependent triggers. Experiments show the method to be effective.

Reviewers unanimously agreed on the method's novelty and efficacy. Initial concerns regarding performance against baselines, ablation studies, and other backbones (like ViT) were well-addressed during the rebuttal. Following the rebuttal, all reviewers were in favor of the paper's acceptance.